# Multilingual Arbitrage: Optimizing Data Pools to Accelerate Multilingual Progress

## Abstract

The use of synthetic data has been crucial in achieving recent state-of-the-art breakthroughs. However, relying solely on a single oracle teacher model for data generation can lead to issues such as model collapse and bias propagation. These problems are particularly pronounced in multilingual contexts, where no single teacher model performs optimally across all languages. In this study, we propose a solution through multilingual arbitrage, which exploits performance variations among multiple models for each language. By strategically routing samples through a diverse set of models, each possessing unique strengths in different languages, we address these challenges. Our extensive experiments with state-of-the-art models demonstrate that our arbitrage techniques significantly enhance performance compared to relying on a single teacher model. Our multilingual arbitrage techniques result in large gains of up to 80% win-rates over state-of-art proprietary and widely adopted open weight models such as Gemma 2, Llama 3.1, Mistral v0.3. These gains, achieved through multilingual arbitrage and averaged across all languages, were most substantial in the less-resourced languages within our pool.

## 1 Introduction

Throughout our lives, we are guided by many teachers, each contributing distinct insights and expertise to our personal and professional growth. For specialized skills, such as becoming a doctor or mastering culinary arts, we seek out experts who provide targeted guidance. In contrast, synthetic data generation often relies on a single teacher model to impart knowledge to a student model. This approach can lead to the passive transfer of both the strengths and limitations inherent in the teacher model, as highlighted in various studies (Shumailov et al., 2023; Magister et al., 2023; Shimabucoro et al., 2024; Gerstgrasser et al., 2024). Moreover, it assumes that a single model can effectively teach all relevant skills, which may not always be the case.

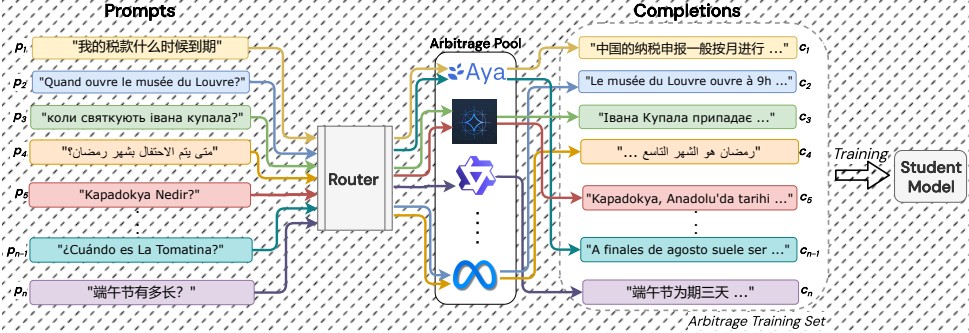

Figure 1: **Overview of *Multilingual Arbitrage*.** Instead of relying on a single "oracle" teacher, multilingual arbitrage re-frames the distillation problem as learning how to optimize sampling for a desired part of the data distribution from an ensemble of teachers.

The limitations of the single oracle approach become particularly pronounced in multilingual settings, where high-performing large language models (LLMs) are often trained predominantly on a

few data-rich languages (Singh et al., 2024; Joshi et al., 2020; Fan et al., 2021). This diverse landscape of multilingual model development has resulted in a variety of models: large-scale models that support multiple languages (Xue et al., 2020; Scao et al., 2022; Shliazhko et al., 2022; Li et al., 2023; Üstün et al., 2024), frontier models with some multilingual capabilities that are not specifically optimized (Armengol-Estapé et al., 2021; Chowdhery et al., 2022; Zhang et al., 2022; Team et al., 2024), and models focused on regional language families (Adelani et al., 2021; Mirzakhalov et al., 2021; Cahyawijaya et al., 2022). As a result, it is often unclear how to determine which model to use to maximize performance for a given language. Relying on a single model can also further amplify disparities in treatments between languages, as models may perform well on some language but not have coverage for others. Performance tends to be critical for the quality of synthetic data, which can enable further progress in those languages by making data more ubiquitous over time (Alaa et al., 2022; Gao et al., 2023; Bukharin & Zhao, 2023; Li et al., 2024; Zhang et al., 2024).

**In this work, we take a wider view of synthetic data generation.** Instead of viewing model distillation as simply transferring data from a single oracle to a student, we reframe the problem within this heterogeneous landscape as learning how to optimize sampling for a desired part of the data distribution from an ensemble of teachers. Multilingual settings serve as an ideal case study for this approach due to the distinct boundaries between languages compared to tasks. We anticipate that our *arbitrage techniques* will enhance performance in scenarios where it is uncommon for a single model to be state-of-the-art across all tasks.

We introduce the concept of *multilingual arbitrage*, which leverages performance differences among models for a given language. For each language, we utilize a pool of models as potential teachers and evaluate strategic sampling methods by routing to different models. This optimized distribution is then used to instruction fine-tune a new multilingual model, aiming to surpass the performance of a single multilingual model across all languages. This approach raises the question: *Can strategic sampling from multiple models outperform individual models?*

We conducted exhaustive experiments across 15 languages using 9 state-of-the-art multilingual models to evaluate our method. Our extensive evaluations included LLM-as-an-evaluator win rates, discriminative tasks, and textual characteristics. Our key findings and contributions are as follows:

- **We introduce the concept of "multilingual arbitrage" which significantly outperforms traditional single teacher distillation.** Our experiments demonstrate that arbitrage methods surpass single-teacher models. Specifically, our reward-based routing technique achieved an average improvement of 56.5% in generative win rates and a 28.1% improvement over the best single-teacher model. Additionally, student models trained using this technique exhibited an average absolute gain in win rates of 32.02% (a relative gain of 153.5%) over various state-of-the-art models, and 6.9% absolute improvement (15.9% relative improvement) over the best model highlighting the significant performance advantage of our approach.

- **Not all arbitrage techniques are equal.** We evaluate the performance of various arbitrage techniques against a lower bound baseline of random routing. Reward-based routing, fixed routing with predefined set of expert teachers, and learned routing improved absolute performance by 30.6%, 22.9% and 13.4% (relative performance by 119.5%, 76.8%, and 40.6%) respectively. While reward-based routing, though resource-intensive, was the most effective, our results show that the more efficient reward-guided learned routing can achieve impressive performance gains without needing to generate all completions from each model.

- **Arbitrage improves or maintains textual characteristics.** We analyze the textual characteristics of student model generations by calculating various statistics scores, examining the effects of instruction fine-tuning (IFT) with multilingual arbitrage on text verbosity, readability, and lexical diversity. Our findings show that reward-based routing results in a 14.1% increase in the number of tokens in generated text, while learned routing leads to a 68.4% increase compared to both single-teacher generations (averaged across all single teachers) and random routing. Additionally, we observe increases in lexical diversity scores: reward-based routing improves scores by 6%, and learned routing by 4.2% compared to single teachers, and by 13.4% and 11.5% compared to random routing, respectively.

- **Arbitrage results in a model checkpoint which outperforms state-of-art models.** We scaled our arbitrage setup and compared it to state-of-the-art models such as Gemma 2 (Team et al., 2024), Llama 3.1 (Dubey et al., 2024), and Mistral v0.3[1]. Specifically, we observed an average absolute gain in win rates of 32.02% (a relative gain of 153.5%) compared to various state-of-the-art models, resulting in absolute win rates for our arbitrage models ranging from 50.1% to 80% against Gemma 2 and Mistral v0.3, respectively.

## 2 METHODOLOGY

Our primary goal is to train a high-performing multilingual student model $S$. Given a set of input prompts $P = \{p_i\}_{i=1}^N$, we generate a corresponding set of completions $C = \{c_i\}_{i=1}^N$ using a pool of potential teacher models $\mathcal{T} = \{T_j\}_{j=1}^M$. These prompt-completion pairs $(p_i, c_i)$ will then be used to fine-tune $S$. For each prompt $p_i \in P$, we aim to identify the specific teacher model $T_j \in \mathcal{T}$ that produces the highest quality completion $c_i$.

We consider that each teacher model $T_j$ may not perform uniformly across all regions of interest $R$ in the data distribution. Therefore, we aim to minimize the empirical error $E[P_j(R)]$, where $P_j(R)$ represents the performance of teacher model $T_j$ in region $R$, over the broader distribution $D$. This ensures robustness and generalization beyond the i.i.d. training sample $D_{\text{iid}}$. This approach allows us to select the most suitable teacher model for each prompt, optimizing the training of our student model $S$. We note that this amounts to optimization in the data space and allows for *on-the-fly* creation of dataset properties to minimize sensitivity to distribution drift.

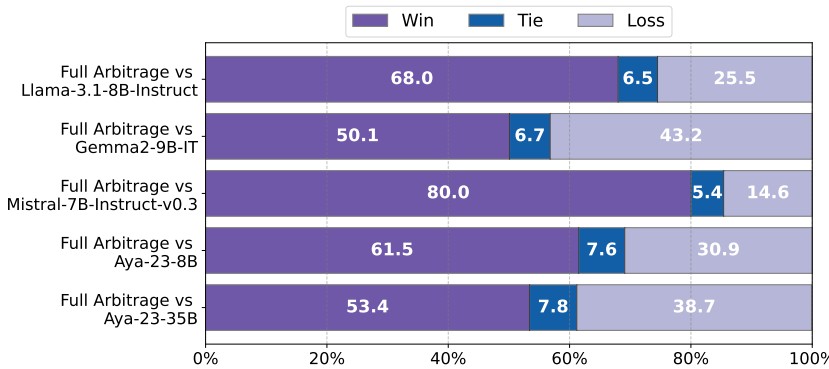

Figure 2: **Win rates (%) of student trained with arbitrage data:** Comparison of reward-based routing trained students with state-of-the-art models. The largest gain is observed with a 65.4% win-loss difference against *Mistral-7B-instruct-v0.3*. Values are aggregated across 23 languages.

### 2.1 ROUTING METHODS

The crux of the problem of multilingual arbitrage is: *how do you route prompts to the most calibrated teacher model for each prompt?* We exhaustively benchmark different routing strategies which we introduce briefly below:

**Fixed Routing.** In practice, one might choose a fixed model, such as $T_2$, to process all input prompts in $P$. This can be reasonable if $T_2$ demonstrates significantly better overall performance for a majority of the prompts. In the multilingual case, this setting is one in which we can pre-determine the best model for each language based on their known strengths, enabling us to use a fixed routing strategy for each prompt deterministically by choosing the appropriate teacher model according to the prompt's language. However, in real-world settings it is not always possible to know what models are relatively strong at different languages in advance.

**Reward-based Routing.** Next we consider the more realistic setting which assumes that we cannot pre-determine a fixed routing strategy. Instead, we rely on a reward model for routing. For each $p_i$ we generate a completion from each of the teacher models in $\mathcal{T}$ and then select $c_i$ to be the

---

[1]https://huggingface.co/mistralai/Mistral-7B-Instruct-v0.3

completion with the highest score given by some ranking method. In our case, we use a proprietary reward model (Cohere May 2024) which is competitive with top-scoring state-of-the-art reward models on the RewardBench Leaderboard (Lambert et al., 2024)[2]. We intentionally use a separate reward model for routing from the model that we use for our LLM-as-a-judge evaluation (GPT-4-Turbo[3]) given the known biases incurred by using the same model for both (Bansal et al., 2023; Verga et al., 2024; Shimabucoro et al., 2024).

**Learned-Routing.** The disadvantage of reward-based routing is that it requires generating a full set of $M$ completions for each prompt where $M = |\mathcal{T}|$. As a more efficient alternative, we explore the merits of a *learned router* which instead trains a router model based on scores produced by the reward model which is proposed by (Lu et al., 2024). In this method, the router model learns to predict the reward conditioned only on the prompt $p_i$, thereby determining the most suitable teacher model $T_j$ without the need to generate multiple completions based upon historical routing trends. The router $R(p_i)$ is defined to select the teacher model $T_j$ that maximizes the expected reward for a given prompt $p_i$. Formally, for each $p_i \in P$, the selected model $T_j$ is given by:

$$T_j = \arg \max_{T \in \mathcal{T}} R(p_i, T).$$

This approach leverages the complementary strengths of the models in $\mathcal{T}$ and ensures that each prompt is routed to the model most likely to produce the highest quality completion. By integrating reward model ranking with query routing, reward-guided Learned-Routing enhances the efficiency of the LLM ensemble, reducing computational overhead while ensuring effective training of the student model $S$.

To train our learned-routing model, we collect a training dataset of diverse prompts and then generate completions from each of the candidate models in the teacher pool. Given a prompt from our training set, we obtain a scalar reward for each candidate model generation as in the following:

$$\mathbf{r_i} = \{RM(p_i, T_j(p_i))\}_{j=1}^{|\mathcal{T}|}, \quad i = 1, \dots, N \tag{1}$$

where $\mathbf{r_i} \in \mathcal{R}^{|\mathcal{T}|}$. We then train our router $R$ on the training data with Kullback-Leibler (KL) divergence as the loss function:

$$\mathcal{L}(p_i, \mathbf{r}_i) = \mathrm{KL}(R(p_i), \mathrm{softmax}(\mathbf{r}_i)). \tag{2}$$

This approach improves the quality of synthetic data while maintaining computational efficiency during inference, introducing only minimal overhead compared to traditional reward model ranking methods, which is training the router model. However, this overhead is well compensated during inference because learned routing only generates samples from the routed model, rather than from each model in the pool. As a result, the generation cost is reduced to $1/M$, where $M$ is the number of models in the pool.

## 3 EXPERIMENTAL SETUP

### 3.1 BASELINES

To evaluate the effectiveness of *multilingual arbitrage*, we compare our method against several baseline methods. Below, we provide a brief overview of the experimental details for each baseline:

**Single Teachers.** This is the most widely adopted framework for incorporating synthetic data into training. In this paradigm a student model is trained on the generations produced from a single teacher model. This setup allows us to explore the question: ***Is multilingual arbitrage more effective than a single "oracle" teacher?***

We choose single teacher models based on their architecture, size, base model type, and language coverage. Our experiments are divided into two scales. For the basic set, we use widely adopted models with parameters ranging from 7B to 9B: Aya 23 (Aryabumi et al., 2024), Llama 3 (Dubey et al., 2024), and Gemma 2 (Team et al., 2024). For larger-scale experiments with expanded language coverage, we choose top-performing open-weight models: CommandR+, Gemma2 27B (Team et al.,

---

[2]https://huggingface.co/spaces/allenai/reward-bench
[3]https://platform.openai.com/docs/models/gpt-4-turbo-and-gpt-4

2024), and Mistral Large 2. Detailed information about each model is provided in Appendix A.2. Although Llama 3 and Gemma 2 do not explicitly claim multilingual support, they are often used by multilingual users more than models explicitly designed for multiple languages, such as mT0 (Muennighoff et al., 2023) and BLOOMZ (Muennighoff et al., 2023).

**Random Routing.** Next, we consider a router that **randomly** assigns each prompt $p_i \in P$ to teacher model $T_j \in \mathcal{T}$, without considering the language or any other specific characteristics of the prompt. This allows us to investigate: *Is multilingual arbitrage better than a random selection as to what model is best for a given distribution of interest?*

**Translation.** Lastly, our translation baseline addresses whether strategic sampling outperforms simply translating the generations of a single English model into multiple languages. We aim to determine: *Does generating synthetic data in the target language outperform translating the best English only data?*

We generate completions for our English training prompts using our most capable English teacher model, Llama 3. We then translate each of the prompts and completions to the seven languages included in our router experiments.

## 3.2 ROUTING TEACHER POOLS

**Fixed Router Model Pool.** In our fixed router experiments, we assume prior knowledge of which models perform best for specific languages. We train several geo-cluster models on 15 languages, each specialized in different language groups: ***Germanic*** which includes German and Dutch; ***Slavic*** consisting of Czech, Russian, Ukrainian, Polish; ***Romance*** covering French, Portuguese, Spanish, Italian, Romanian); and ***East-Asian*** consisting of Turkish in addition to Korean, Japanese, Chinese. This allows models to exploit geographic and linguistic similarities within a language cluster (Kohli et al., 2023; Kew et al., 2023; Tejaswi et al., 2024). Each geo-cluster outperforms the single teacher model before student model training, achieving an average absolute win rate gain of 5.95% (relative gain of 14.9%) over single teachers. Additional training and win rate evaluation details are provided in Appendix A.2.1.

**Reward-based and Learned Routing.** These methods aims to demonstrate the effectiveness of routing in a varied pool of models with unknown multilingual performance. Hence, we consider a diverse pool that includes all single teacher models (3.1), geo-cluster models (3.2) and monolingual models in Chinese (Qwen2-7B-instruct (Yang et al., 2024)) and Turkish (Turkish-Llama-8b-Instruct-v0.1) which are specifically designed to support individual languages. We include more details about the monolingual models in Appendix A.2. This variety, ranging from massively multilingual to geo-cluster and monolingual models, helps us analyze which types of models are most utilized by different routing techniques.

**Learned Routing** To train our learned router, we fine-tune the Gemma2-2B(Team et al., 2024) model, selected for its compact size, strong performance, and multilingual capabilities. To further improve training efficiency, we also evaluate a smaller mT5-base (Xue et al., 2020) variant with 580M parameters. Comparative results for these models are presented in Appendix A.4. Our learned router models were trained using prompts from Dolly-15k which were translated using NLLB-3.3B (Team et al., 2022) into the seven languages covered by our routing experiments, and resulting in 60,419 prompts in total.

## 3.3 STUDENT MODEL

We chose the Aya 23 8B model (Aryabumi et al., 2024) as our student model due to its state-of-the-art multilingual capabilities for its size. Our experiments are conducted at two scales: i) ***Basic Set*** where synthetic data is generated in seven languages: *Arabic, Chinese, English, French, German, Turkish*, and *Ukrainian* and ii) ***Larger Scale*** where synthetic data is generated in 23 languages, including the initial seven plus: *Dutch, Czech, Greek, Spanish, Persian, French, Hebrew, Hindi, Indonesian, Italian, Japanese, Korean, Polish, Portuguese, Russian*, and *Vietnamese*. These languages cover diverse language families to ensure comprehensive evaluation across various linguistic contexts (see Table 6 in Appendix A.3).

**Training Details.** For the basic set, student models are trained using 10,000 randomly sampled prompts from the *UltraFeedback Binarized Dataset* (UFB) (Tunstall et al., 2023), an English pref-

erence dataset with 61,135 pairs. These prompts are translated into seven target languages using the NLLB-3.3B model, resulting in 70,000 prompts. For larger-scale experiments, 10,000 UFB prompts, 13,000 from Dolly (Conover et al., 2023), and 43,000 from ShareGPT [4] are translated into 23 languages, totaling 1,358,000 prompts. Completions for each prompt are generated by the assigned teacher model. Each student model is then instruction fine-tuned on these multilingual data points - 70,000 for the basic set and 1,358,000 for the larger scale—selected through multilingual arbitrage.

The training employed a cosine learning rate schedule with a warm-up phase, using a batch size of 64 and an evaluation batch size of 128. The peak learning rate was set at $2.5 \times 10^{-5}$, achieved through 128 warm-up steps starting from a learning rate of $0.0$, and then decayed back to $0.0$.

### 3.4 Evaluations

**Open-ended Generation Win rates.** Beyond traditional NLP tasks, we aim to evaluate the open-ended generation capabilities of the student models, focusing on their ability to produce unstructured and long-form responses. For this evaluation, we use GPT-4 as an LLM-judge to measure pairwise win rates between two model generations. We evaluate on the target language subset of the Multilingual Dolly-200 Eval dataset (Singh et al., 2024; Üstün et al., 2024). This 200 instance evaluation dataset is a held-out curated sample from the Dolly-15k dataset (Conover et al., 2023). These prompts are open-ended and capture general-purpose non-code use cases. Hence, evaluation using this dataset is a valuable proxy for how multilingual arbitrage impacts more fluid and often open-ended asks.

**Discriminative Tasks.** To evaluate our models on completely unseen tasks, we follow Muennighoff et al. (2023) and use XNLI (Conneau et al., 2018), XCOPA (Ponti et al., 2020), and XStoryCloze (Lin et al., 2021) datasets targeting natural language inference, commonsense reasoning and sentence completion respectively. These unseen tasks are crucial for evaluating the effectiveness of IFT in improving a model's reasoning and comprehension capabilities as they test the model's ability to discriminate between different possible interpretations or outcomes. For all unseen tasks, we report zero-shot performance.

## 4 Results and Discussion

### 4.1 Multilingual Arbitrage Performance

**Comparison against state-of-the-art models.** Figure 2 shows the win rates of our reward-based arbitrage routing strategy compared to several widely adopted models, with parameters ranging from 7B to 9B, as well as the Aya 23 model with 35B parameters. Our student models, trained using data derived from this strategy, demonstrated a significant performance advantage over all these state-of-the-art models. We observed an average absolute increase in win rates of 32.02% (relative gain of 153.5%) across all models, with improvements ranging from 6.9% (15.9% relative) for Gemma2 9B to 65.4% (447% relative) for Mistral-7B-instruct, based on results averaged across 23 languages.

**Comparison against random routing.** Our random routing baseline serves as a crucial lower bound that any proposed arbitrage strategy should outperform. This baseline helps us evaluate: *Is our multilingual arbitrage technique better than a random guess?* In Figure 3, we compare the win rates of each of the different routing methods against the random routing baseline. We observe that all the multilingual arbitrage methods consistently outperformed the random baseline with average win rate of 51.8% and a notable absolute win rate improvement of 22.3% (78.9% relative) on average.

**Comparison against single "oracle" teacher.** In Figure 4, we show win rates comparing our arbitrage routing strategies to single teacher models. Student models trained with data from these strategies significantly outperformed those using single teacher generations. Specifically, fixed routing achieves an absolute average winrate improvement of 13.3% (34.7% relative), reward-based routing shows a 19.5% absolute average improvement (56.5% relative), and learned routing has a 9.0% absolute improvement in average (25.6% relative) over all single teachers. Notably, Gemma 2

---

[4]https://sharegpt.com/

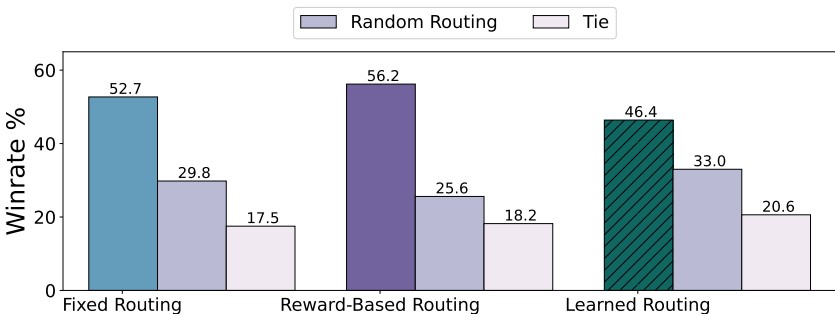

Figure 3: **Win rates (%) of students trained with different routing strategies:** Comparison of router-trained and random routing trained students. Reward-based routing shows the largest gains with a 30.6% win-loss difference. Values are percentages aggregated across 7 languages.

was the best-performing single teacher, yet learned routing still achieved an absolute average winrate improvement of 1.4% (3.2% relative gain) over it.

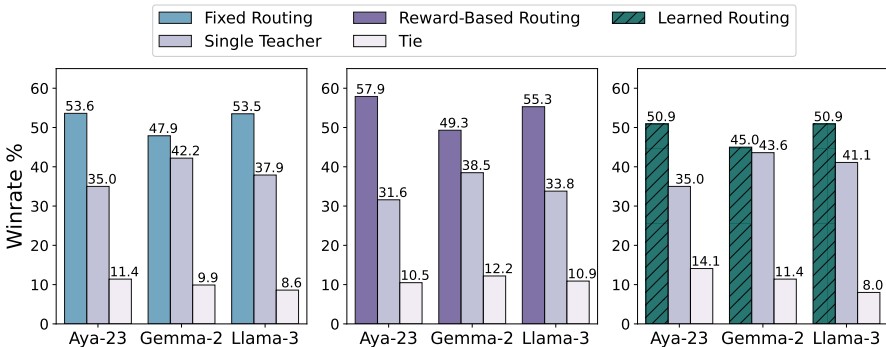

Figure 4: **Win rate (%) comparison of Fixed Routing, Reward-Based Routing and Learned Routing against Single Teacher Models.** The x-axis shows the single teacher model used for synthetic data generation. All multilingual arbitrage strategies outperform single teachers, with reward-based routing achieving the largest gains. Values are aggregated across seven languages: *Arabic, Chinese, English, French, German, Turkish, and Ukrainian.*

**Win-rate Gains are largest for Reward-Based Routing.** We observed the largest improvements against single teachers with reward-based routing, achieving average gains of 56.5%. However, reward-based routing is the least efficient arbitrage method because it requires running inference and generating completions with all models in the pool for each prompt. Although fixed routing and learned routing show some decrease in win-rates compared to reward-based routing, they are significantly more efficient during inference, as they only require inference from one model. In our experiments with a pool of 9 models, reward-based routing requires generating and scoring 9 completions per prompt, while fixed and learned routing need only one generation per prompt. Although learned routing involves an additional call to the router per prompt, this router model is much smaller and more efficient than the teacher, making the call negligible compared to generating from all models in the pool. Notably, learned routing is the most flexible technique, being 9 times more efficient than reward-based routing in this setup and not needing prior knowledge of each model's merits, unlike fixed routing.

**Discriminative tasks.** Table 8 presents average scores for unseen discriminative tasks, reporting zero-shot performance. These tasks reveal similar gaps between the benefits of single teachers and arbitrage techniques. Single teachers provide an average absolute improvement of 0.57 (0.98% relative improvement) over the base student model (Aya 23), while arbitrage techniques achieve a larger absolute average improvement of 1.14 (1.95% relative improvement).

Overall, on discriminative tasks, Fixed Routing emerges as the most effective, with the highest absolute average improvement of 1.46 (2.50% relative) across tasks, followed by reward-based routing

| | XCOPA | XNLI | XStoryCloze | Average |
|---|---|---|---|---|
| AYA23 (Base Model) | 64.1 | 42.9 | 68.23 | 58.41 |
| SINGLE TEACHERS | 65.5 | 43.96 | 67.41 | 58.98 ↑ 0.98 |
| RANDOM ROUTING | 65.9 | 44.01 | 67.25 | 59.05 ↑ 1.09 |
| FIXED ROUTING | **67.4** | 43.89 | 68.33 | **59.87** ↑ 2.50 |
| REWARD BASED ROUTING | 66.2 | **44.21** | 68.20 | 59.53 ↑ 1.91 |
| LEARNED ROUTER | 65.8 | 43.62 | **68.36** | 59.25 ↑ 1.43 |

Table 1: **Performance of Student Models on held-out Discriminative Tasks:** XCOPA, XNLI, and XStoryCloze. Results are averaged over seven languages, showing performance changes relative to the base model AYA23. Single teacher results are averaged across Aya23, Llama 3, and Gemma 2. The 'Average' column includes the percentage increase over the base model.

with a 1.12 (1.91% relative) improvement, indicating their superior ability to enhance cross-lingual and commonsense reasoning capabilities in the student models. Notably, while fixed routing ranks first in discriminative tasks, it is second in win rate comparisons. This discrepancy may stem from a noted tension between model performance on academic benchmarks and open-ended generations. Recent studies suggest that as performance on open-ended tasks improves, traditional academic task performance may decline (Iyer et al., 2023; Üstün et al., 2024). This occurs because supervised fine-tuning of large language models has increasingly been torn between objectives: improving traditional academic benchmarks and training LLMs to follow instructions, acquire conversational abilities, and be helpful and harmless (Aakanksha et al., 2024). See Table 8 in Appendix A.6 for comprehensive results.

## 4.2 LANGUAGE AND ROUTING ANALYSIS

**Difference in per-language gains.** Figure 5 shows performance gains for medium- versus high-resource languages when using reward-based and learned routing strategies compared to single teacher models such as Aya 23, Llama 3, and Gemma 2.

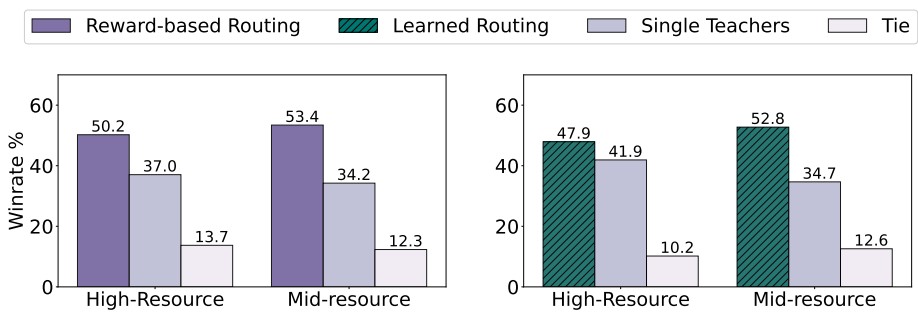

Figure 5: **Win rate Changes Across Language Resource Level:** Comparison of the Mid-Resource Languages and High-Resource Languages win rates against Single Teachers (results are the average of Aya 23, Llama 3 and Gemma 2 single teachers). Mid-resource languages consist of Turkish and Ukrainian and high-resource languages are English, German, French, Chinese and Arabic.

Medium-resource languages, Turkish and Ukrainian, experience greater benefits, with reward-based routing achieving an absolute gain of 19.2%(56.1% relative gain) and learned routing achieving a 18.1% (52.2% relative gain) over single teachers. In contrast, high-resource languages (Joshi et al., 2020), English, German, French, Chinese, and Arabic see an absolute gain of 13.2% (35.7% relative gain) with reward-based routing and 6% (14.3% relative gain) with learned routing. These findings suggest that medium-resource languages gain more from routing strategies than from single teacher models. Detailed per-language gains are provided in Table 7 in Appendix A.5.

**Routed Dataset Distribution Across Models.** In Figure 6, we illustrate the distribution of the training dataset prompts routed to each model (for the reward-based router). We observed a balanced routing strategy with different models favored for each language, which highlights the benefits of combining the strengths of a pool of models with varying strengths. For instance, Llama 3, a strong English model, receives 60% of English prompts but is less frequently used for other languages. Meanwhile, 30.7% of Chinese prompts are directed to the Chinese monolingual expert, whereas the Turkish monolingual expert is rarely selected, with only 0.6% of prompts routed to it. Overall, Aya 23 emerges as the leading multilingual model, predominantly chosen for Ukrainian, Turkish, and Arabic, with 53% of Arabic prompts routed to it. Geo-cluster models, included for all languages except Arabic (as there is no Geo-cluster model for it), handle an average of 18.7% of the prompts.

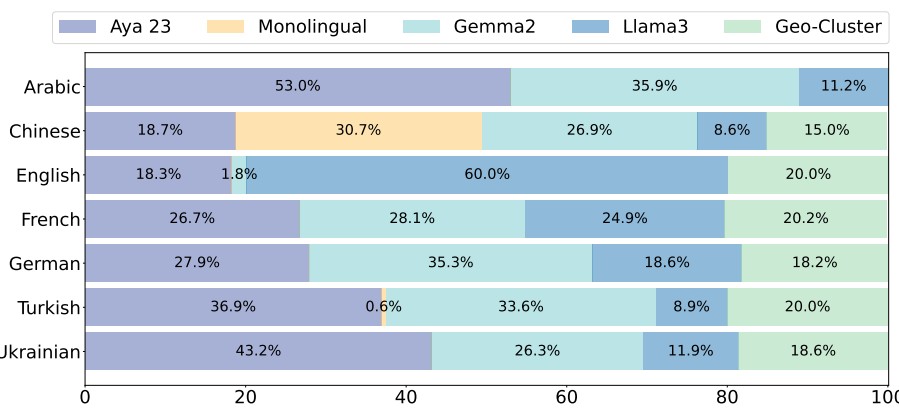

Figure 6: **Model Composition per Language:** Here we analyze the model routing distribution of a dataset constructed with Reward-Based Routing. The values represent the percentage of prompts routed to a given model for the particular language.

**Comparison of in-language generation vs translation.** In this section, we explore whether generating synthetic data directly in the target language is more effective than translating the best English-only data. To investigate this, we first generate English data using Llama 3 (the best English model), translate it into other 6 languages, and train a student model with this translated data. We then compare this student model's performance to those trained with Llama 3's single-teacher generations and random-routing generations.

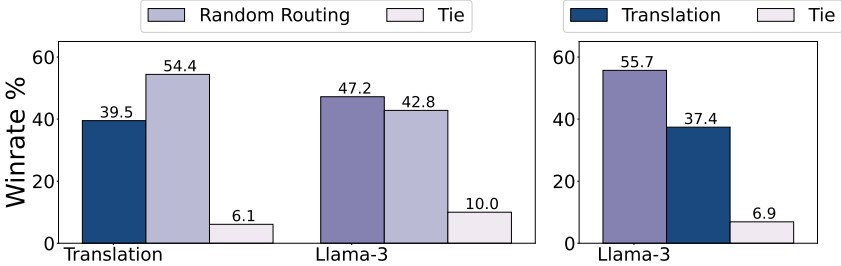

Figure 7: **Win rates (%) of students trained with Llama 3 translations and Llama 3 generations:** Comparison of translation, in-language generation by single teacher and router-trained students to those trained with random routing. The largest gains are observed for in-language data generation with a win-loss diff of 18.3%. All values are aggregated over 7 languages.

Figure 7 demonstrates that random routing outperforms the translation baseline, achieving a win rate of 54.4%, while the Llama 3 single teacher model exceeds the random-routing baseline with 4.4% (10.3% relative) gain in this experiment. Direct comparison of Llama 3 translation with Llama 3 single teacher students exhibits a significant absolute 18.3 % (48.9% relative) increase in win rates for the single teacher model. These results indicate that translation is the least effective method for synthetic data generation, as even random routing performs better. Generating samples within the original language offers substantial advantages over relying on single model translations, despite the single model being the top performer in the original language (English) before translation to others.

### 4.3 TEXTUAL CHARACTERISTICS

To gain a holistic view of how multilingual arbitrage affects model generation characteristics, we use the TextDescriptives framework from Hansen et al. (2023) to calculate various textual features. We report average statistics, including token count, readability, and lexical diversity scores. Detailed analyses of the textual characteristics of generations are provided in Appendix A.7.

## 5 RELATED WORK

**LLM circularity.** The issue of LLM circularity, where models influence others through distilled data, has gained attention, focusing on model degradation and self-preference (Dohmatob et al., 2024; Briesch et al., 2023; Shumailov et al., 2023). Recursive training impairs performance by neglecting long-tail knowledge (Briesch et al., 2023; Bertrand et al., 2024; Shumailov et al., 2024), leading to a loss of diversity (Guo et al., 2024; Feng et al., 2024). (Shimabucoro et al., 2024) explore how the transfer of characteristics via passive inheritance occurs when synthetic data generated by different LLMs is involved. By considering the issues highlighted in these studies, we aim to optimize synthetic data generation by selecting the most calibrated teacher model from a pool of LLMs in a multilingual setting.

**Instruction Fine-tuning (IFT) and Multilingual Synthetic Data.** IFT enhances LLM performance and generalization (Sanh et al., 2021; Wei et al., 2021; Mishra et al., 2021; Min et al., 2021; Ouyang et al., 2022), relying on task diversity (Longpre et al., 2023; Wang et al., 2023b; Chung et al., 2022), complexity (Xu et al., 2023; Luo et al., 2023), and quality (Zhou et al., 2023; Taori et al., 2023). While validated mainly for English tasks, there is a growing focus on multilingual contexts (Üstün et al., 2024). Efforts address multilingual instruction dataset scarcity (Singh et al., 2024). Research on English synthetic data generation is extensive (Gao et al., 2023; Anaby-Tavor et al., 2019), but its multilingual impact is less understood (Kaddour & Liu, 2023). Recent studies explore multilingual data with a single teacher model (Aryabumi et al., 2024) and for preference training (Aakanksha et al., 2024). In this work, we strategically sample from a diverse pool of models, each with unique strengths across different languages, to generate high-quality synthetic instruction data. Our research diverges by concentrating on multilingual synthetic instruction data generation from an ecosystem view rather than a single teacher.

**Large Language Model Ensemble.** Ensembling LLMs leverages individual strengths, but limited research exists on these effective strategies. Frameworks combine LLMs using pairwise ranking and generative fusion (Jiang et al., 2023), sequential inference (Chen et al., 2023), and supervised learning for output fusion (Wang et al., 2023a). Routers select the best LLM candidate based on benchmarks (Shnitzer et al., 2023). Relevant work proposes reward model-guided routing for task strengths (Lu et al., 2024). Our work explores various routing strategies beyond reward-based routing, in multilingual contexts.

## 6 CONCLUSION

In this work, we introduce the concept of *multilingual arbitrage*, which strategically utilizes performance variations across different models for a given language to sample from a pool of teacher models, thereby generating a superior dataset for training effective student models. Our extensive experiments across 23 languages demonstrate the efficacy of our routing strategies, significantly enhancing student models' performance across all benchmarks. We observed notable gains in both open-ended generation tasks and discriminative benchmarks compared to the traditional single-teacher data generation and training method. Furthermore, additional analysis of textual characteristics and evaluation on unseen discriminative tasks confirm that our instruction fine-tuned students not only retain their initial capabilities but also improve their multilingual generation skills. Our findings underscore the value of strategic sampling, particularly in scenarios where a diverse pool of models can excel at different parts of the data distribution. We expect *arbitrage* techniques will yield substantial gains in addressing out-of-distribution challenges and in handling rare or underrepresented parts of the data distribution.

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

# A  APPENDIX

## A.1  OVERVIEW OF ARBITRAGE TECHNIQUES

|  | Fixed | Reward-Based | Learned |
|---|---|---|---|
| Works with Unknown Teachers | ✗ | ✔ | ✔ |
| All models are considered for each prompt | ✗ | ✔ | ✔ |
| Efficient Routing | ✔ | ✗ | ✔ |
| New models can be added on-the-fly | ✗ | ✔ | ✗ |

Table 2: **A comparison of different arbitrage techniques:** We compare the properties of the different proposed routing methods. While the reward-based routing is the most flexible approach, it comes at the cost of efficiency as compared to the learned router.

## A.2  TEACHER MODEL POOL DETAILS

**Single Teacher Models.** We include additional details about each of the single teacher models we benchmark below:

- **Aya-23-8B** (Aryabumi et al., 2024) is an 8B parameter model and a part of the Aya-23 family of multilingual instruction-tuned language models that supports 23 languages, and are based on Cohere's Command model[5] and multilingual instruction-style collection (Singh et al., 2024).

- **Llama-3-8B-instruct** (Dubey et al., 2024) is an open-source instruction-tuned version of the Llama-3-8B pre-trained model. The model is trained on over 15 trillion tokens of publicly available data, with a focus on optimizing the performance across various real-world scenarios, including reasoning and code generation.

- **Gemma-2-9B-it** (Team et al., 2024) is a 9B parameter instruction fine-tuned model on 8T tokens of data from web documents, code, and science articles. In particular, the 9B model was trained with knowledge distillation (Hinton et al., 2015) instead of next token prediction.

- **Gemma-2-27B-it** (Team et al., 2024) is a 27B parameter instruction fine-tuned model on 13T tokens of data from web documents, code, mathematics.

- **Command-r-plus-08-2024** [6] is a 104B parameter multilingual model optimized for 10 languages: English, French, Spanish, Italian, German, Brazilian Portuguese, Japanese, Korean, Arabic, and Simplified Chinese.

- **Mistral Large 2**[7] is a 123B parameter instruction fine-tuned model, supports dozens of languages including French, German, Spanish, Italian, Portuguese, Arabic, Hindi, Russian, Chinese, Japanese, and Korean.

**Monolingual Teacher Models.** These models are specifically tailored for individual languages, specifically Chinese and Turkish:

- **Qwen2-7B-instruct** (Yang et al., 2024) is an open-source 7B parameter model pretrained on 7T tokens of data from code, mathematics, and multilingual data. Qwen2-7B-instruct is a multilingual model supporting approximately 30 languages, and showing strong performance on Chinese.

---

[5] https://cohere.com/command
[6] https://huggingface.co/CohereForAI/c4ai-command-r-plus
[7] https://huggingface.co/mistralai/Mistral-Large-Instruct-2407

- **Turkish-Llama-8b-Instruct-v0.1**[8] is a fully fine-tuned version of the Llama-3-8B-instruct model with a 30GB Turkish dataset. It currently tops the Turkish leaderboard on Hugging-Face[9] for text generation tasks.

### A.2.1 GEO-CLUSTER TRAINING DETAILS

| Language Cluster | Languages |
|---|---|
| GERMANIC | German, Dutch |
| SLAVIC | Czech, Russian, Ukrainian, Polish |
| ROMANCE | French, Portuguese, Spanish, Italian, Romanian |
| EAST-ASIAN | Korean, Japanese, Chinese, Turkish |

Table 3: **Language composition of Geo-clusters:** To evaluate fixed routing, we control apriori for the strength of a model on each language in our pool by training Geo-cluster models which are specialized on different groups of languages.

To train highly performant Geo-clusters, we train an 8B parameter Cohere command model on a data mix of the 15 languages covered by the Geo-Clusters as shown in Table 3.

| | Number of Samples Per Dataset | | | |
|---|---|---|---|---|
| Language Cluster | Original ShareGPT | ShareGPT CommandR+ | Original Dolly15k | Dolly15k CommandR+ |
| GERMANIC | 155,480 | 157,699 | 40,466 | 42,447 |
| SLAVIC | 259,217 | 263,488 | 67,721 | 71,121 |
| ROMANCE | 309,708 | 314,513 | 80,295 | 84,345 |
| EAST ASIAN | 230,848 | 235,369 | 58,864 | 61,743 |

Table 4: Number of Training Samples Per Language Cluster

For this data mix, we used both ShareGPT dataset and the Dolly-15k dataset as described by (Aryabumi et al., 2024). First these two datasets' prompts and completions were translated into these 15 languages, and translations were done using the NLLB-3.3B model (Costa-jussà et al., 2022). In addition, we also included what we call the ShareGPT CommandR+ dataset and the Dolly-15k CommandR+ dataset. For these variants, we use the translated prompts generated completions for the translated prompts using Command R+[10]. Our datasets cover 15 languages shown in Table 3. Table 4 shows the training data distribution in terms of number of samples used for each Geo-Cluster model training.

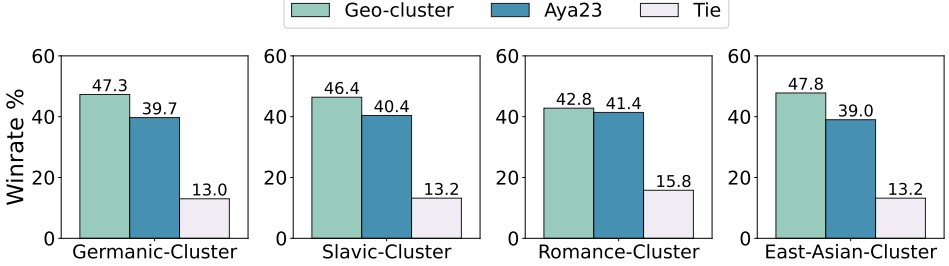

Figure 8: Geo-cluster win-rates against Aya 23 Single Teacher Model after training. All values are percentages, and aggregated over number of languages in each language cluster. Geo-cluster are powerful teacher models relative to the capabilities of the base Aya model.

---

[8]https://huggingface.co/ytu-ce-cosmos/Turkish-Llama-8b-Instruct-v0.1
[9]https://huggingface.co/spaces/malhajar/OpenLLMTurkishLeaderboard_v0.2
[10]https://huggingface.co/CohereForAI/c4ai-command-r-plus

Before using the geo-clusters as teacher models, we validate performance of our trained Geo-cluster models. We compute average win rates in each language cluster using the held-out multilingual Dolly-200 evaluation dataset (Üstün et al., 2024).

| Language | Model Pool |
|----------|-----------|
| ARABIC | Base Pool |
| CHINESE | Base Pool, East Asian + Turkish Cluster, Qwen2-7B |
| ENGLISH | Base Pool, Germanic Cluster |
| FRENCH | Base Pool, Romance Cluster |
| GERMAN | Base Pool, Germanic Cluster |
| TURKISH | Base Pool, East Asian + Turkish Cluster, Turkish-Llama-8b |
| UKRAINIAN | Base Pool, Slavic Cluster |

Table 5: Teacher model pool available for each language. The *Base Pool* consists of those outlined in Section 3.1: Aya 23, Llama 3, Gemma 2.

## A.3 LANGUAGE FAMILIES

As we present in Section 3.3, we generate synthetic data in seven diverse languages: *Arabic, Chinese, English, French, German, Turkish, Ukrainian, Dutch, Czech, Greek, Spanish, Persian, French, Hebrew, Hindi, Indonesian, Italian, Japanese, Korean, Polish, Portuguese, Russian, Vietnamese.* These languages, representing different language families, are selected to ensure a comprehensive evaluation across various linguistic contexts, detailed in Table 6.

| ISO Code | Language | Script | Family | Subgrouping | Resources |
|----------|----------|--------|--------|-------------|-----------|
| ara | Arabic | Arabic | Afro-Asiatic | Semitic | High |
| zho | Chinese | Han | Sino-Tibetan | Sinitic | High |
| eng | English | Latin | Indo-European | Germanic | High |
| fra | French | Latin | Indo-European | Italic | High |
| deu | German | Latin | Indo-European | Germanic | High |
| tur | Turkish | Latin | Turkic | Common Turkic | Mid |
| ukr | Ukrainian | Cyrillic | Indo-European | Balto-Slavic | Mid |
| nld | Dutch | Latin | Indo-European | Germanic | High |
| ces | Czech | Latin | Indo-European | Balto-Slavic | High |
| ell | Greek | Greek | Indo-European | Graeco-Phrygian | Mid |
| spa | Spanish | Latin | Indo-European | Italic | High |
| pes | Persian | Arabic | Indo-European | Iranian | High |
| fra | French | Latin | Indo-European | Italic | High |
| heb | Hebrew | Hebrew | Afro-Asiatic | Semitic | Mid |
| hin | Hindi | Devanagari | Indo-European | Indo-Aryan | High |
| ind | Indonesian | Latin | Austronesian | Malayo-Polynesian | Mid |
| ita | Italian | Latin | Indo-European | Italic | High |
| jpn | Japanese | Japanese | Japonic | Japanesic | High |
| kor | Korean | Hangul | Koreanic | Korean | Mid |
| pol | Polish | Latin | Indo-European | Balto-Slavic | High |
| por | Portuguese | Latin | Indo-European | Italic | High |
| rus | Russian | Cyrillic | Indo-European | Balto-Slavic | High |
| vie | Vietnamese | Latin | Austroasiatic | Vietic | High |

Table 6: **Lineage for Cluster Languages.** 23 languages covered by our main experiments, each language's corresponding script, family, subgrouping, and if it is classified as higher or mid-resourced according to (Joshi et al., 2020).

### A.4 ROUTER MODEL DETAILS

**Training Details.** We chose Gemma2-2B[11] as our router model for its compact size, performance, and multilingual capabilities. We fine-tuned Gemma2-2B model using the AdamW (Loshchilov & Hutter, 2019) optimizer with an initial learning rate of $5 \times 10^{-5}$. We used a linear learning rate scheduler with a 200 warmup steps. We set weight decay to 0 and fine-tuned for 2 epochs.

To further improve training efficiency, we also evaluate a smaller mT5-base[12] variant with 580M parameters. We finetuned the mT5-base using the Adafactor optimizer with $1 \times 10^{-3}$ as the learning rate. We fine-tuned for 5 epochs with a train batch size of 32.

**Comparison of mT5 and Gemma 2 as Router Model.** We chose Gemma2-2B as the final candidate for our learned router model. The student model trained on the dataset routed by Gemma2-2B demonstrated significant improvements, particularly against the strong Gemma2-9B single teacher model. Gemma2-2B was used as the learned router in all our experiments.

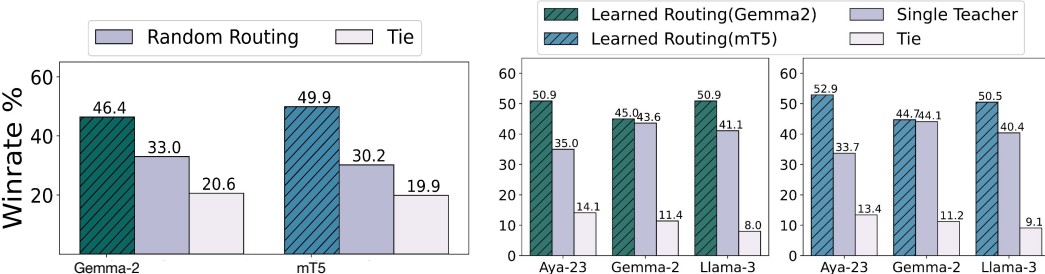

Figure 9: Win-rate % comparison of Learned Routing (mT5) and Learned Routing (Gemma2) against Random Routing (left) and multiple Single Teacher Models (right).

Figure 9 shows Gemma2-2B and mT5-base router performances compared to random routing and single teachers. Despite its smaller size, mT5-base also achieved remarkable results, outperforming all baseline approaches with a notable 65.2% gain over random routing and an average gain of 27.7% over single teacher models.

### A.5 DIFFERENCE IN PER-LANGUAGE GAINS.

In Figure 10, we compare both reward-based routing and learned routing strategies against random routing for medium-resource and high-resource languages.

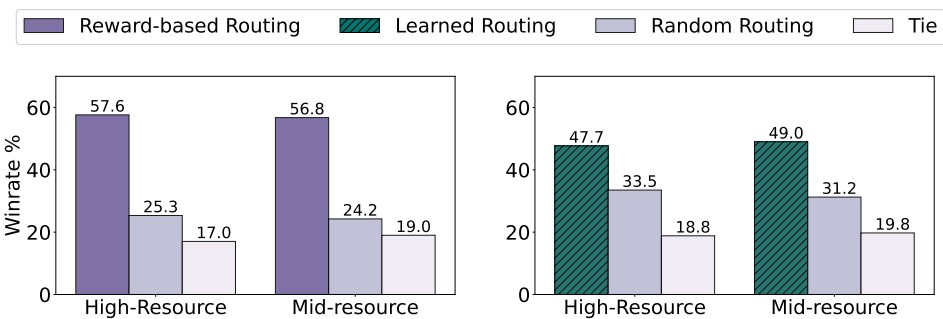

Figure 10: **Win-rate Changes Across Language Resource Level.** We compare the win rates of Mid-Resource Languages and High-Resource Languages against random-routing. Mid-resource languages consist of Turkish and Ukrainian and high-resource languages are English, German, French, Chinese and Arabic.

---

[11]https://huggingface.co/google/gemma-2-2b

[12]https://huggingface.co/google/mt5-base

High-resource languages (Joshi et al., 2020), English, German, French, Chinese, and Arabic see a 127.6% gain with reward-based routing and a 42.4% gain with learned routing. Medium-resource languages that includes Turkish and Ukrainian, experience greater benefits, with reward-based routing achieving a 134.7% gain and learned routing achieving a 57.1% gain over random routing. These findings suggest that medium-resource languages gain more from strategic sampling than from random routing. Detailed per-language gains are provided in Table 7.

| Language | % gain (Single Teachers) | | % gain (Random Routing) | |
|---|---|---|---|---|
| | **Reward-based** | **Learned** | **Reward-based** | **Learned** |
| Arabic | 75.7 | 43.4 | 115.1 | 43.5 |
| Chinese | **114.5** | 2.9 | 101.8 | -4.6 |
| English | 55.2 | 0.4 | 116.0 | **115.7** |
| French | 22.5 | -4.4 | 79.3 | 39.1 |
| German | 31.7 | 28.8 | 76.7 | 88.7 |
| Turkish | 52.2 | **59.6** | **228.9** | 94.5 |
| Ukrainian | 59.9 | 43.7 | 172.9 | 87.2 |

Table 7: **Win-rate gains across languages.** This table presents the percentage gain of reward-based routing and learned routing compared to single teachers and random routing across seven languages. The highest gain in each column is highlighted in **bold**, while the second highest gain is indicated in blue.

The results indicate that reward-based routing leads to larger gains across all languages compared to learned routing, whether against single teachers or random routing. Mid-resource languages, Turkish and Ukrainian, consistently show high gains in all scenarios, followed by Arabic. However, no distinct pattern emerges for high-resource languages. Notably, reward-based routing results in significant gains for Chinese against both random routing and single teachers. Additionally, both reward-based and learned routing achieve substantial gains for English when compared to random routing.

## A.6 Discriminative tasks.

|  | XCOPA | XNLI | XStoryCloze | Average |
|---|---|---|---|---|
| **Base Model** |  |  |  |  |
| AYA23 (Base) | 64.1 | 42.9 | 68.23 | 58.41 |
| **Single Teacher** |  |  |  |  |
| AYA23 | 65.5 ↑ 2.18 | 43.86 ↑ 2.23 | 68.05 ↓ 0.27 | 59.13 ↑ 1.23 |
| LLAMA-3 | 65.1 ↑ 1.56 | 44.04 ↑ 2.65 | 66.46 ↓ 2.60 | 58.53 ↑ 0.20 |
| GEMMA-2 | 66.1 ↑ 3.12 | 43.98 ↑ 2.51 | 67.74 ↓ 0.72 | 59.3 ↑ 1.52 |
| Translation | 64.6 ↑ 0.78 | 43.46 ↑ 1.30 | 66.77 ↓ 2.14 | 58.27 ↓ 0.24 |
| **Multilingual Arbitrage** |  |  |  |  |
| Random Routing | 65.9 ↑ 2.80 | 44.01 ↑ 2.58 | 67.25 ↓ 1.44 | 59.05 ↑ 1.09 |
| Fixed Routing | **67.4** ↑ 5.14 | 43.89 ↑ 2.30 | 68.33 ↑ 0.14 | **59.87** ↑ 2.50 |
| Reward Based Routing | 66.2 ↑ 3.27 | **44.21** ↑ 3.05 | 68.20 ↓ 0.05 | 59.53 ↑ 1.91 |
| Learned Router | 65.8 ↑ 2.65 | 43.62 ↑ 1.67 | **68.36** ↑ 0.19 | 59.25 ↑ 1.43 |

Table 8: Performance of Student Models on held-out Discriminative Tasks: XCOPA, XNLI, and XStoryCloze. The results are averaged over seven languages, highlighting the improvements or declines in performance compared to the base model AYA23.

## A.7 Textual Characteristics

To obtain a more holistic view of how multilingual arbitrage impacts model generation characteristics, we utilize the TextDescriptives framework from Hansen et al. (2023) to calculate various textual features. We report average statistics, including the number of tokens along with readability and lexical diversity scores. Metrics like length are straightforward to compute and serve as positively correlated proxies for quality (Singh et al., 2024). These metrics are calculated from model generations over 100 instances from the Dolly200 Eval set (Singh et al., 2024). We standardize comparisons across models by allowing for a maximum output length of 600 tokens.

| Student Models | # Tokens | Gunning-Fog | Rix | MLTD |
|---|---|---|---|---|
| AYA23 (Base) | 76.74 | 15.83 | 4.7 | 43.98 |
| **Single Teacher Students** |  |  |  |  |
| AYA23 | 151.83 | 17.67 | 5.92 | 46.51 |
| LLAMA-3 | 141.71 | 17.33 | 5.87 | 49.5 |
| GEMMA-2 | 140.59 | 15.67 ↓ | 4.28 ↓ | 52.48 |
| Translation | 197.05 | 16.62 | 5.22 | **53.01** |
| **Multilingual Arbitrage** |  |  |  |  |
| Random Routing | 144.16 | 17.16 | 5.81 | 45.81 |
| Fixed Routing | 160.75 | 17.71 | 5.94 | 50.79 |
| Reward Based Routing | 164.4 | 17.01 | 5.69 | 51.95 |
| Learned Routing | **242.56** | **19.11** | **7.74** | 51.08 |

Table 9: Evaluation of textual characteristics across student models in 4 languages: English, German, French and Ukranian. The number of tokens, Gunning-Fog Index, Rix Index, and Measure of Textual Lexical Diversity (MLTD) for each model highlights the differences in verbosity, readability and lexical diversity. Except for Gemma 2, all students show increase for all metrics.

In addition to basic statistics like length, we also compute:

1. **Gunning Fog Index** (Gunning, 1968) is a readability test that estimates the years of formal education required to understand a piece of text on the first reading. Gunning-Fog uses sentence length and prevalence of complex words to estimate the complexity of the text and assign a grade level between 0 and 20. A score of 17-18 indicates college graduate-level text.

2. **Rix** (Anderson, 1983) calculates readability based on the number of words with more than six characters divided by the number of sentences in the text. A score of 5 corresponds to a grade level of around 10, while a score of 7 or higher indicates the need for a higher educational level to comprehend.

3. **Measure of Textual Lexical Diversity (MTLD) score** (Shen, 2022) helps tracking changes in vocabulary by reflecting the average number of words in a sequence that maintains a certain type-token ratio (TTR), a measure of vocabulary variety (McCarthy & Jarvis, 2010). An MLTD score of 50 can be considered as moderate lexical diversity.

All the results are presented in Table 9 and Figure 11.

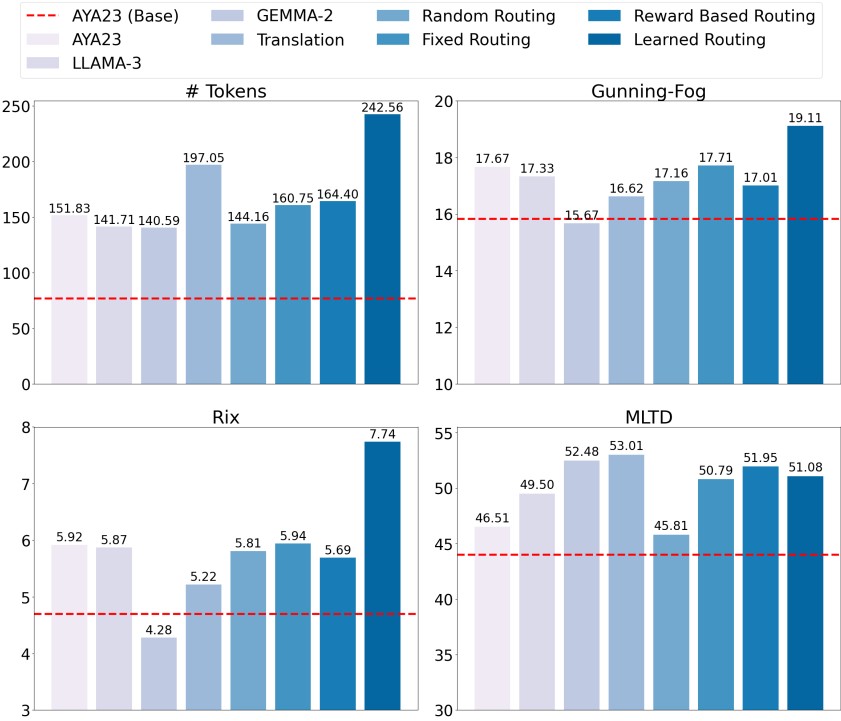

Figure 11: **Evaluation of Textual Characteristics:** We analyze characteristics of student models in four languages: ENGLISH, GERMAN, FRENCH AND UKRANIAN. The number of tokens, Gunning-Fog, Rix Index, and MLTD for each model highlights the differences in verbosity, readability and lexical diversity.

**Average number of tokens per generation.** The most significant change is observed in the average number of tokens per generation. The base model generates an average of 76 tokens per generation, whereas routing approaches produce substantially longer outputs, ranging from 160 tokens with Fixed Routing to 242 tokens with Learned Routing. In contrast, both random routing and single teacher models (averaged across Aya 23, Llama 3, and Gemma 2) generate around 144 tokens on average. These findings demonstrate that arbitrage methods result in longer text generations compared to both random routing and single teacher models.

**Textual properties.** The readability metrics show smaller absolute changes compared to the average number of tokens. For the Gunning-Fog index, changes range from a decrease of 0.16 for Gemma

2 to an increase of 3.28 for Learned Routing, relative to the base student model. Similarly, the Rix index varies from a decrease of 0.42 for Gemma 2 to an increase of 3.04 for Learned Routing. Both metrics reveal that arbitrage methods result in higher scores. The Gunning-Fog index shows an absolute difference of 1.05 between arbitrage methods and single teacher models, whereas the difference is 0.78 for random routing. For the Rix index, the absolute difference is 1.11 between arbitrage methods and single teachers, compared to 0.65 for random routing.

These indices serve as proxies for evaluating text complexity. There is a clear trend indicating that multilingual arbitrage strategies, especially the learned routing approach, lead to higher readability metrics. In contrast, single teacher models, especially Gemma 2, generally result in lower values.

Regarding the MLTD score, we observe significant changes, with Reward-based routing showing an increase of up to 7.97 and Learned routing showing an increase of 7.1 relative to the base student model, which are considered substantial improvements (Treffers-Daller et al., 2016). Arbitrage methods result in higher MLTD scores compared to both random routing and single teacher results. The average absolute difference is 1.77 between arbitrage methods (averaged over all 3 methods) and single teacher models (averaged over Aya 23, Llama 3 and Gemma 2), while the difference is 5.46 for random routing.

Overall, multilingual arbitrage strategies significantly increase the number of tokens in generations, readability metrics and improve lexical diversity compared to single teacher models. This suggests that multilingual arbitrage enhances data quality and diversity, which in turn leads to improvements in student model performance and explains the significant increase in win rates.

**Routed Dataset Composition Characteristics.** Here, we analyze how prompt characteristics affect the reward-based router decision, using the same subset of the UltraFeedback Binarized Dataset (UFB) as depicted in Figure 6. The average MLTD score and number of tokens of the prompts routed to a particular model is shown in Figure 12.

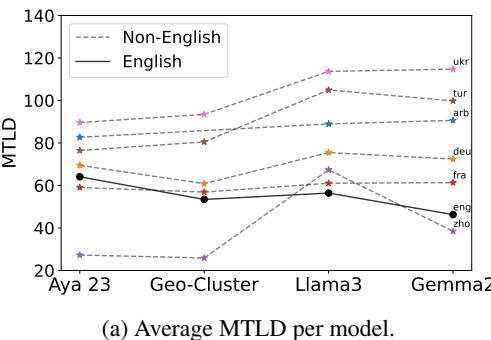
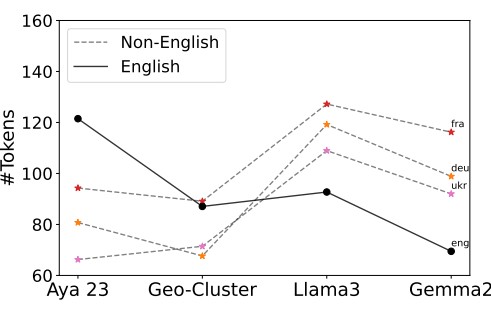

(a) Average MTLD per model.      (b) Average number of tokens per model.

Figure 12: **Characteristics of Prompts Routed to Given Models:** We analyze the MTLD (a) and number of tokens (b) for the set of prompts routed to each of the teacher models as selected by Reward-Based Routing. Each line represents a different language and each column is a particular teacher model.

Figure 12a shows that the average MTLD scores *for English prompts* routed to different models range from 46.28 to 64.07. Aya 23 receives English prompts with the highest MLTD score of 64.07, while Llama 3 has an average MLTD score of 56.41, and Gemma 2 has the lowest score of 46.28. In contrast, for non-English prompts, Aya 23 has an average MLTD score of 67.42, Llama 3 scores 79.66, and Gemma 2 achieves the highest MLTD score of 85.24.

Figure 12b shows that the longest English prompts are routed to Aya 23, with an average of 121.5 tokens, while Gemma 2 receives the shortest English prompts, averaging 69.4 tokens. English prompts routed to Geo-clusters and Llama 3 have average token counts of 87.1 and 92.7, respectively. *For non-English prompts, the pattern differs.* Geo-clusters receive the shortest prompts, averaging 78.8 tokens. Aya 23 receives prompts with an average of 90.7 tokens, Gemma 2 with 94.1 tokens, and Llama 3 receives the longest non-English prompts, averaging 112.0 tokens.

We can conclude, for English prompts, those that are more lexically diverse and longer tend to be routed to Aya 23. In contrast, for non-English prompts, Gemma 2 and Llama 3 are preferred for handling more lexically diverse and longer prompts.

## A.8 FULL BUDGET COMPARISON

To show the effectiveness of reward-based routing, we also compare it against a variant, we refer to as *Full Budget*. In this variant, we include the completions generated by all $M$ teacher models in the pool for each prompt. This results in a dataset with $M$ times more data points than the other variants presented in the paper. The results demonstrate that strategic sampling outperforms even the version where all generations from all models are used.

| Language | Reward-Based Routing | All Completions | Tie |
|---|---|---|---|
| ENGLISH | **54.0** | 31.5 | 14.5 |
| GERMAN | **47.5** | 33.5 | 19.0 |
| FRENCH | **50.0** | 34.0 | 16.0 |
| ARABIC | **46.5** | 34.5 | 19.0 |
| CHINESE | **51.0** | 39.0 | 10.0 |
| TURKISH | **54.5** | 27.5 | 18.0 |
| UKRAINIAN | **45.0** | 34.0 | 21.0 |

Table 10: **Win rates (%) Comparison** of Reward-based routing trained student with all completions trained student model. The Reward-based routing variant consistently outperforms the latter with the highest gain in Turkish.

## A.9 LANGUAGE-SPECIFIC WIN RATES

We present the language-specific win rates (%) for 23 languages, comparing the *Reward-Based Routing* model against the best-performing state-of-the-art model in our experiments, *Gemma2-9B-IT*. In 19 of these languages, the model trained with the reward-based routing approach achieves higher win rates than *Gemma2-9B-IT*.

| Language Code | Reward-Based Routing | Gemma2-9B-IT | Tie |
|---|---|---|---|
| ar | **57.5** | 36.0 | 6.5 |
| cs | **50.5** | 42.5 | 7.0 |
| de | **50.0** | 46.0 | 4.0 |
| el | **57.0** | 37.5 | 5.5 |
| en | 37.0 | **57.0** | 6.0 |
| es | 41.0 | **52.5** | 6.5 |
| fa | **57.0** | 35.5 | 7.5 |
| fr | 35.0 | **55.5** | 9.5 |
| he | **65.0** | 29.0 | 6.0 |
| hi | **50.0** | 44.0 | 6.0 |
| id | **53.0** | 42.0 | 5.0 |
| it | **49.5** | 42.0 | 8.5 |
| ja | 44.0 | **45.5** | 10.5 |
| ko | **53.0** | 34.5 | 12.5 |
| nl | **49.0** | 46.0 | 5.0 |
| pl | **49.0** | 45.0 | 6.0 |
| pt | **46.5** | 45.5 | 8.0 |
| ro | **58.0** | 37.5 | 4.5 |
| ru | **50.5** | 44.5 | 5.0 |
| tr | **47.5** | 45.0 | 7.5 |
| uk | **48.5** | 44.5 | 7.0 |
| vi | **53.5** | 42.5 | 4.0 |
| zh | **50.5** | 44.5 | 5.0 |

Table 11: **Language-Specific Win Rates (%)** of Reward-Based Routing Compared to Gemma2-9B-IT Across 23 Languages.

A.10   STUDENT MODEL GENERATIONS

Examples of model generations in English from different student models can be found in Table 12.

| | |
|---|---|
| **Prompt:** *Cats have relatively few taste buds compared to humans (470 or so versus more than 9,000 on the human tongue). Domestic and wild cats share a taste receptor gene mutation that keeps their sweet taste buds from binding to sugary molecules, leaving them with no ability to taste sweetness. Their taste buds instead respond to acids, amino acids like protein, and bitter tastes. Cats also have a distinct temperature preference for their food, preferring food with a temperature around 38 °C (100 °F) which is similar to that of a fresh kill and routinely rejecting food presented cold or refrigerated (which would signal to the cat that the "prey" item is long dead and therefore possibly toxic or decomposing). Why are cats so picky about what they eat?* | |

| Student Model | Generation |
|---|---|
| Single Llama 3 | Cats are known for being picky eaters, and their unique taste preferences are due to a combination of factors. The main reason is that cats have a specific temperature preference for their food, which is around 38°C (100°F). This temperature is similar to the temperature of a fresh kill, which is around 37°C (98°F) to 40°C (104°F). Cats are also born with a natural instinct to hunt and kill prey, and they are trained to associate certain temperatures with food that is fresh and safe to eat. In addition, cats have a limited number of taste buds compared to humans, which is around 470 taste buds on the tongue. This is because cats have a unique taste receptor gene mutation that prevents their sweet taste buds from binding to sugary molecules. Instead, their taste buds respond to acids, amino acids, and bitter tastes. This means that cats are not as sensitive to sweet tastes as humans are, and they are more likely to be attracted to foods that have a strong flavor or aroma. Another factor that contributes to cats' picky eating habits is their ability to detect and avoid potential toxins. Cats have a natural instinct to avoid food that is toxic or decomposed, and they are trained to associate certain temperatures and textures with potential toxins. This is why cats often reject food that is presented cold or refrigerated, as it may indicate that the food is no longer fresh or safe to eat. Overall, cats' picky eating habits are a result of their unique biological and behavioral characteristics, including their temperature preference, limited number of taste buds, and ability to detect and avoid potential toxins. |
| Random Routing | They have a distinct temperature preference for their food. |
| Reward-Based Routing | According to the text, cats are picky about what they eat because they have a distinct temperature preference for their food. They prefer food with a temperature around 38 °C (100 °F), which is similar to that of a fresh kill and routinely reject food presented cold or refrigerated. This is because cats have a genetic mutation that prevents their sweet taste buds from binding to sugary molecules, leaving them with no ability to taste sweetness. Their taste buds instead respond to acids, amino acids like protein, and bitter tastes. |
| Learned Routing | Cats are picky eaters due to several factors related to their biological and evolutionary traits: **Taste Sensitivity:** Cats have fewer taste buds than humans and cannot taste sweetness due to a genetic mutation. Their taste buds are more responsive to acids, amino acids, and bitter tastes, aligning with their carnivorous diet. **Temperature Preference:** Cats prefer food at around 38°C (100°F), similar to a fresh kill, and often reject cold or refrigerated food, which they may perceive as old and potentially unsafe. These factors ensure that cats consume food that meets their nutritional needs and is safe, reflecting their evolutionary adaptations as obligate carnivores. |

Table 12: Comparison of student model responses trained using different methods: Single Llama 3 (best English teacher), random routing, reward-based routing, and learned routing. The Single Llama 3 student's answer is too long and includes a lot of irrelevant details, while random routing generates responses that are too brief and insufficient.

Examples of model generations in Turkish from different student models can be found in Table 13.

**Prompt:** *Kedilerin, insanlara kıyasla nispeten az sayıda tat alma tomurcuğu vardır (insan dilindeki 9.000'den fazla ile karşılaştırıldığında yaklaşık 470). Evcil ve yabani kediler tat alıcı gen mutasyonunu paylaşıyor. Bu da tatlı tat alma tomurcuklarının şekerli moleküllere bağlanmasını engelliyor ve tatlı tadı alma yeteneklerini kaybediyor. Onun yerine tat alma tomurcukları asitlere, protein gibi amino asitlere ve acı tatlara tepki verir. Kediler ayrıca yiyecekleri için belirgin bir sıcaklık tercihine sahiptir, taze bir öldürme ile benzer olan yaklaşık 38 °C (100 °F) sıcaklığındaki yiyecekleri tercih eder ve rutin olarak soğuk veya buzdolabında sunulan yiyecekleri reddeder (ki bu, "av" öğesinin uzun zamandır ölü olduğunu ve bu nedenle muhtemelen toksik veya çürümeye başladığını kedilere işaret eder). Neden kediler yemek konusunda bu kadar seçicidir?*

| Student Model | Generation |
|---|---|
| Single Aya 23 | Kediler, yiyecekleri için belirgin bir sıcaklık tercihine sahiptir ve taze bir avın sıcaklığını tercih ederler. |
| Random Routing | Kediler, yiyecekleri için belirgin bir sıcaklık tercihine sahiptir ve taze bir öldürme ile benzer olan yaklaşık 38 °C (100 °F) sıcaklığındaki yiyecekleri tercih ederler. Bu, kedilerin yiyecekleri için belirli bir sıcaklık aralığına sahip oldukları ve bu sıcaklıkta yiyecekleri tercih ettikleri anlamına gelir. Kediler ayrıca soğuk veya buzdolabında sunulan yiyecekleri reddederler, bu da yiyecekleri için belirli bir sıcaklık aralığına sahip oldukları ve bu sıcaklıkta yiyecekleri tercih ettikleri anlamına gelir. Bu, kedilerin yiyecekleri için belirli bir sıcaklık aralığına sahip oldukları ve bu sıcaklıkta yiyecekleri tercih ettikleri anlamına gelir. |
| Reward-Based Routing | Kediler, yiyecekleri için belirgin bir sıcaklık tercihine sahiptir ve taze bir öldürme ile benzer olan yaklaşık 38 °C (100 °F) sıcaklığındaki yiyecekleri tercih ederler. Bu, kedilerin yiyecekleri için bir sıcaklık tercihine sahip oldukları ve bu sıcaklıkta yiyecekleri tercih ettikleri anlamına gelir. Kediler ayrıca soğuk veya buzdolabında sunulan yiyecekleri reddederler, bu da yiyeceklerin "av" öğesinin uzun zamandır ölü olduğunu ve bu nedenle muhtemelen toksik veya çürümeye başladığını kedilere işaret ettiği anlamına gelir. Bu nedenle, kediler yiyecek konusunda bu kadar seçici olabilirler. |
| Learned Routing | Kediler, yiyecekleri için belirgin bir sıcaklık tercihine sahiptir ve taze bir öldürme ile benzer olan yaklaşık 38 °C (100 °F) sıcaklığındaki yiyecekleri tercih ederler. Bu, yiyeceklerin taze ve toksik olmamasını garanti eder. |

Table 13: Comparison of student model responses trained using different methods: Single Aya 23 (best Turkish teacher), random routing, reward-based routing, and learned routing. The Aya 23 student's answer is too short and inadequate, while random-routing generates responses that are repetitive.

