# OpenReview forum: "Multilingual Arbitrage: Optimizing Data Pools to Accelerate Multilingual Progress"
_ICLR.cc/2025/Conference — Submitted to ICLR 2025_

### Official Review · Reviewer_87gS · 2024-11-03

**Soundness:** 2
**Presentation:** 3
**Contribution:** 3
**Rating:** 5
**Confidence:** 4

**Summary:**

This paper explores strategies to make use of different teacher LLMs for synthetic data generation for fine-tuning student LLMs. In specific, the authors propose to use:
- fixed routing with prior information about the strength of each teacher
- reward-based routing: a reward model selects the best output among all teacher outputs.
- a learned router: it predicts the most suitable teacher based on the above information on the fly.
These methods are then compared with using a single teacher and randomly routing to different teachers.

The methods are tested on open-ended generations measured by LLM-as-a-judge win rates and some more structured tests measured by accuracy. All three proposed methods outperform a single teacher or random routing. Specifically, in open-ended scenarios: reward-based routing > fixed routing > learned routing; and on multilingual benchmarks the results are mixed. The paper then included three types of analyses which are informative.

**Strengths:**

1. I support the idea of "multilingual arbitrage" (although not necessarily the naming): it is interesting and important to study methods that can use different teacher models to produce synthetic data for different tasks, languages, or even prompts. I think the three methods are also reasonable.
2. The experimentation is well-structured and extensive with informative analysis components.

**Weaknesses:**

1. Score reporting: In many places, it's misleading to compare/report win rates (%) using relative percentages. In my opinion, it's better for these to be presented as absolute differences. For example, when WR improves from 10% to 20%, the improvement should be 10% instead of 100%.
2. Evaluation: All test sets used for evaluation are "translated multilingual", which is also the nature of the prompts/questions used for fine-tuning the LLMs and the learned router. Even worse, the dolly-200 test and the model training data are translated using the same model NLLB-3.3.B. Perhaps the authors can report some results on the test split of the Aya prompts?
3. Although the reward-based routing uses a different reward model/judge than the evaluation, the judges may still correlate in some way. This makes reward-based data synthesis reward hacking-prone. Another minor thing is that the results seem a bit "expected", because reward-based routing uses more resources than fixed routing, which is also more advantageous (having information about each teacher model) than random/single-teacher routing.

**Questions:**

1. Perhaps incorrect citations? BLOOMZ (Dac Lai et al., 2023), Dolly-15k dataset (Costa-juss` a et al., 2022).
2. Regarding the selection of teacher and student LLMs:
    - For the "basic set" experiment, is one of the cases using Aya-23-8B as both the teacher and the student in the "single teacher" setting?
    - In other routing cases, sometimes Aya-23-8B would also be used to synthesize data for its own fine-tuning. This seems weird to me.

---

> ### Author Response · Authors · 2024-11-21
> **Score reporting**
>
> >**Score reporting: In many places, it's misleading to compare/report win rates (%) using relative percentages. In my opinion, it's better for these to be presented as absolute differences. For example, when WR improves from 10% to 20%, the improvement should be 10% instead of 100%.**
>
> Thank you for raising this point. We acknowledge the importance of distinguishing between absolute and relative percentages when reporting win rates. In our work, we chose to present both metrics because they highlight different aspects of improvement. While the absolute difference (e.g., a WR increase from 10% to 20% is a 10% improvement) is already shown in the figures and tables, we also discuss relative improvement (e.g., a 100% increase in this case) in the text to emphasize its significance, particularly in multilingual or underrepresented group settings where even small absolute gains can represent substantial progress. Our intention is to provide a comprehensive view by reporting both measures. If any part of the manuscript lacks clarity on this approach, we are happy to revise it to improve the submission.

---

> ### Author Response · Authors · 2024-11-21
> **Evaluation set**
>
> >**All test sets used for evaluation are "translated multilingual", which is also the nature of the prompts/questions used for fine-tuning the LLMs and the learned router. Even worse, the dolly-200 test and the model training data are translated using the same model NLLB-3.3.B. Perhaps the authors can report some results on the test split of the Aya prompts?**
>
> Thank you for this valuable suggestion. We would like to clarify a few points regarding the evaluation setup. While the prompts are indeed translated using the NLLB-3.3B model, the completions used during training are model-generated, resulting in a mix of translated prompts and generated completions. Additionally, our evaluation employs multiple benchmarks and metrics, and models trained with arbitrage strategies consistently outperform their counterparts across these benchmarks. Importantly, all methods are evaluated on the same datasets, ensuring a fair apples-to-apples comparison between our proposed methods and baselines.
>
> To address your concern further, we tested our models on the Aya evaluation suite (https://huggingface.co/datasets/CohereForAI/aya_evaluation_suite),  where prompts were post-edited by human annotators after translation. This ensures that the dataset is free from any potential biases introduced by automated translation models. Below, we present results in the same setting as Figure 2 for Arabic, Chinese, Portuguese, and Turkish:
>
> | Language | Full Arbitrage | Gemma2-9B-IT | Tie |
> |:-------------:|:-------------:|:-------------:|:-------------:|
> | ar | 52.5 | 37.8 | 9.7 |
> | pt | 46.7 | 44.2 | 9.1 |
> | tr | 49.1 | 42.6 | 8.3 |
> | zh | 45.2 | 43.9 | 10.9 |
>
> These results demonstrate that our model performs effectively on human-edited datasets as well, confirming that our findings hold across both translated and human-edited evaluation sets. We will include the results on the whole evaluation set in the revised version of the paper.

---

> ### Author Response · Authors · 2024-11-21
> **Reward model and judge model correlation**
>
> > **Although the reward-based routing uses a different reward model/judge than the evaluation, the judges may still correlate in some way. This makes reward-based data synthesis reward hacking-prone. Another minor thing is that the results seem a bit "expected", because reward-based routing uses more resources than fixed routing, which is also more advantageous (having information about each teacher model) than random/single-teacher routing.**
>
> To mitigate this risk, we have taken steps to separate the reward distribution observed during routing from the judge decisions during evaluation. As mentioned in Section 2.1, the reward model used for routing (Cohere May 2024) and the evaluation judge (GPT-4-Turbo) are distinct. This separation was intentional to avoid biases that can arise when the same model is used for both purposes [2,3]. So far, we have not observed or found clear evidence of correlation between these two models. However, we are open to suggestions on how to further evaluate potential correlations between the reward model and the evaluation judge.
>
> We do not consider "Fixed Routing" and "Reward-based Routing" as direct competitors, as they serve different use cases. Reward-based routing is resource-intensive but delivers the best performance when there are no budget constraints. On the other hand, fixed routing is more cost-effective and can be preferred when prior knowledge about the models is available. In fixed routing, we leverage prior information to assign prompts to specific models, for instance, the Germanic geo-cluster specializes in languages like German and Dutch etc. In this setup, German prompts are routed to the Germanic cluster and Ukrainian prompts to the Slavic cluster for data generation. While this approach works well with strong prior knowledge, it is often impractical in real-world scenarios where such information is unavailable.
>
> We hope this addresses your concerns. Please feel free to reach out if you have any further questions or feedback.
>
> [2] Verga et al, "Replacing Judges with Juries: Evaluating LLM Generations with a Panel of Diverse Models.", arXiv:2404.18796, 2024.
>
> [3] Shimabucoro et al, "LLM See, LLM Do: Guiding Data Generation to Target Non-Differentiable Objectives", EMNLP, 2024.

---

> ### Author Response · Authors · 2024-11-21
> **Citations**
>
> > **Perhaps incorrect citations? BLOOMZ (Dac Lai et al., 2023), Dolly-15k dataset (Costa-juss` a et al., 2022).**
>
> Many thanks for bringing this to our attention. We apologize for the mistake and have already corrected this in the paper.

---

> ### Author Response · Authors · 2024-11-21
> **Aya 23 as both teacher and student model**
>
> >**For the "basic set" experiment, is one of the cases using Aya-23-8B as both the teacher and the student in the "single teacher" setting?**
>
> >**In other routing cases, sometimes Aya-23-8B would also be used to synthesize data for its own fine-tuning. This seems weird to me.**
>
> Thank you for highlighting this point. We selected the Aya 23 8B model as both a student and one of the teacher models due to its state-of-the-art multilingual capabilities relative to its size. While it is common practice to use a model’s own generations to enhance performance through self-distillation [2,3], it’s important to clarify that Aya 23 8B is not the sole teacher model used for synthetic data generation in our experiments.
>
> As shown in Figure 6, approximately 32% of prompts in the base setting are routed to Aya 23 8B, reflecting its role among a pool of teacher models. In larger settings, where we employ nine different teacher models, even fewer prompts are routed to Aya 23 8B, further diversifying the sources of knowledge.
>
> Additionally, we use a completely distinct dataset for distillation that has never been used for training or fine-tuning the student model. This ensures that the student model can still learn from its own generations, when prompts are routed to Aya 23 as the teacher, as well as from those of other teacher models, rather than simply reinforcing pre-existing knowledge.
>
> [4] Wang, Yizhong, et al. "Self-instruct: Aligning language models with self-generated instructions." arXiv preprint arXiv:2212.10560 (2022).
>
> [5] Yang, Zhaorui, et al. "Self-distillation bridges distribution gap in language model fine-tuning." arXiv preprint arXiv:2402.13669 (2024).

---

> ### Author Response · Authors · 2024-11-24
> **Discussion period is coming to an end**
>
> We truly appreciate your thoughtful feedback and the time you’ve taken to evaluate our manuscript. As the discussion phase is nearing its conclusion, we kindly encourage you to review our responses to your comments. We hope our clarifications address your concerns, but please don’t hesitate to let us know if there are any remaining issues or additional details we can provide to assist in resolving them.

---

> ### Comment · Area_Chair_X27n · 2024-11-25
>
> Dear Reviewer 87gS,
>
> Thank you for your valuable contributions to the review process for the paper! The authors have submitted their rebuttal, and I would greatly appreciate it if you could take a look and provide your response.

---

> ### Comment · Reviewer_87gS · 2024-11-26
> **Reviewer's comment after author response.**
>
> Dear authors, thank you for the responses and additional experimental results.
>
> - **Writing/Content**:
>     - **citations**---addressed.
>     - **score reporting**:
>         - I do prefer absolute percentages because 1) the win rate (WR) itself is already a percentage; 2) by using WR, we ultimately want to measure model quality, yet the scale for WR may not linearly correspond to "quality". If I would like to present a "quality change" from 1% WR to 10% WR, would you agree it's a 9% or 900% quality improvement?
>     - **Title "arbitrage"**
>         - when choosing data using a reward model or running the router, the teacher response quality difference is not exploited and the student model does not benefit from this "arbitrage". There is only "selection" but no "buying and selling". This has also been mentioned by Reviewer Ee5t.
> ----
> - **Experimental setup**:
>     - **New results on the human-edited Aya test**---thanks, addressed
>     - **Aya-23-8B used as both teacher and student**---addressed
>     - **RM and LLM-as-a-judge correlation**
>         - the proposed "reward-based routing" is essentially "data selection using reward models." I think such exploitation is vulnerable to over-optimization to the metric.

---

> ### Author Response · Authors · 2024-11-28
>
> Thank you for taking the time to review our responses. Below we will do our best to clarify the the remaining concerns:
>
> **Score reporting** - We changed the main score reporting to absolute differences in the revision and submitted. You can see these changes now.
>
> **Title arbitrage** - We understand your concern, but it is not allowed to change the title under submission. We will address this concern in case of acceptance and change the title for camera-ready version.
>
> **RM and LLM-as-a-judge correlation** - Could you please clarify this concern further?
>
> To address potential bias, we have ensured a clear separation between the reward distribution observed during routing and the judge's decisions during evaluation. As outlined in Section 2.1, the reward model used for routing (Cohere May 2024) is distinct from the evaluation judge (GPT-4-Turbo). This distinction ensures that the reward model's decisions are independently assessed by a judge from a different model family.
>
> Additionally, our evaluations are supported by academic benchmarks such as **XCOPA**, **XNLI**, and **XStoryCloze**, where we have consistently observed performance improvements.

---

> ### Comment · Reviewer_87gS · 2024-11-30
>
> RMs and LLM-as-a-judges are designed to evaluate model outputs instead of being used in model training. Involving them in the training data selection is essentially optimizing a model to produce outputs liked by a reward model, making it reward-hacking prone.
>
> **RM and LLM-as-a-judge correlation** means that an RM and an LLM-as-a-judge could score the same set of candidates higher than some other candidates, irrespective of their actual quality judged by human annotators. Even though the RM and the LLM judge in this paper are different foundation models, their pre-training data still can overlap significantly.
>
> Before training a student model, if one has to use all teachers to generate an output for all input prompts, and then select instance-level teachers, I think this is **not really routing** but just training data selection using a reward model. Therefore I am worried about the correlation mentioned above without human evaluation.
>
> This, in fact, brings up another interesting question: why does learned routing under-perform fixed routing given that reward-based routing can outperform fixed routing? If the learned router has been trained to perfectly classify the "best" teacher model based on the input only, then it could yield the same performance as the reward-based router. Is it because of a lack of generalizability from Dolly to other training prompts? Is it because that *the best teacher actually cannot be predicted unless their generations are seen*? <- if so, this certainly implies a certain level of overfitting.
>
> ----
>
> Regarding the results on XCOPA, XNLI, and XStoryCloze, they do seem "mixed" to me as noted in the original review. Also, in terms of average scores and improvements, the relative percentage change of accuracy does not make sense---authors should consider absolute accuracy change. Accuracy is already a percentage defined as `num_correct/num_total`.

---

> > ### Author Response · Authors · 2024-12-04
> >
> > Thank you for engaging with the discussion and for raising this question, which allows us to clarify these important points further.
> >
> > We would like to start with the question:
> >
> > >"Why does learned routing under-perform fixed routing given that reward-based routing can outperform fixed routing".
> >
> > Fixed routing benefits from prior knowledge about which model works best for specific languages. For example, Germanic geo-clusters are specialized for languages like German and Dutch, while Slavic geo-clusters are optimized for Czech, Russian, and Ukrainian. Routing based on such strong priors inherently outperforms learned routing because it leverages explicit geographic and linguistic similarities. However, such priors are not always available, making learned routing a more practical and adaptable solution.
> >
> > Reward-based routing improves performance by selecting the teacher model that maximizes rewards for each prompt, but this approach requires generating outputs from all teacher models for every sample, which is computationally expensive. To address this, we proposed a learned router (e.g., mT5-base or Gemma2-2B) trained on historical routing trends and reward scores. This router predicts rewards efficiently without generating multiple outputs per prompt. While its performance may not fully match reward-based routing due to its approximation nature, it achieves a practical balance between efficiency and effectiveness, significantly improving scalability in resource-constrained scenarios.
> >
> > Regarding the comment on:
> > >"if one has to use all teachers to generate an output for all input prompts, and then select instance-level teachers, I think this is not really routing but just training data selection using a reward model."**
> >
> > We leave the choice to practitioners. If budget permits, reward-based routing is optimal; otherwise, learned routing offers a cost-effective alternative with better performance than single-teacher or random routing. Notably, our learned routers are purposefully small and trained on only ~10K samples, leaving room for improvement with larger models.
> >
> > Lastly, reward-based routing remains a routing strategy because, even with all teacher generations available, it selects a single model for each prompt based on rewards. As demonstrated in our "Full Budget" experiments, reward-based routing achieves better performance with fewer data and reduced training time compared to using all teacher models simultaneously.

---

### Official Review · Reviewer_su5q · 2024-11-04

**Soundness:** 2
**Presentation:** 2
**Contribution:** 1
**Rating:** 3
**Confidence:** 4

**Summary:**

The paper introduces "mulgilingual arbitrage", a novel distillation method from state-of-the-art LLMs to a smaller student model, by using a router to pick the best teacher model for each input training sample. The student model after such training outperforms its untrained version, together with several other open LLMs. The paper also discusses the difference between three routing stratgies in performance and shows the "reward-based routing" performs the best.

**Strengths:**

1. The method is simple and effective, which can save teacher model inference cost and enhance student model performance.
2. The method can be easily applied to other teacher and student models.

**Weaknesses:**

1. The idea that we can use multiple teacher models to train a student model is rather trivial and intuitive. As a result, the contribution of this paper is relatively small to the research community.
2. Also, the method, training a router to pick the best teacher for each sample, is quite straightforward, while the optimality of which cannot be guaranteed. Actually, the result that "reward-based routing" (using the score from an open reward model to choose the best from all the teachers' responses) outperforms "learned routing" (using a trained router) shows that the trained router does't always choose the teacher with the highest reward for that sample. Furthermore, the reward model score can also carry bias and error, bringing error accumulation in the process, but no experiment in this paper can tell whether the teacher with the highest reward is the ground-truth-best teacher for that sample (reviewed by human experts, etc.).
3. The paper doesn't tell whether the "multilingual arbitrage" method is better than simply mixing the responses from multiple teacher models. Though there is a experiment comparing the reward-based routing method with the "single 'oracle' teacher" method (which is merely using the teacher model with the highest average performance), it is not enough.
4. Some of the experiment settings are questionable, for example:
   - Aya-23-8B is used as both the student model and one of the teacher models
   - Part of the Dolly-15k dataset is used in both the router training set and the system testing set
5. The title and abstract of this paper put "multilingual" as a key point. However, throughout the methodology part, no specific design for the multilingual problem is proposed. Also, in the experment setting part, only 3 discriminative tasks are metioned to test the multilingual capacity of the trained student model, which is not satisfactory. In fact, the routing method can be applied to much more generation tasks, as long as there is need to distil open LLMs into a smaller model.

**Questions:**

1. Why is the router training loss the KL divergence instead of the normal cross-entropy? Since the goal of the router is to choose the single "best" teacher for a given sample, the cross-entropy loss against the highest-reward teacher of a given training sample seems more reasonable.

---

> ### Author Response · Authors · 2024-11-21
> **Contribution of the paper**
>
> We would like to thank reviewer [**su5q**] for their invaluable feedback and for recognizing the method as "simple and effective" while also acknowledging its adaptability to other teacher and student models. Below, we address some of their concerns:
>
> > **The idea that we can use multiple teacher models to train a student model is rather trivial and intuitive. As a result, the contribution of this paper is relatively small to the research community**
>
> Thank you for raising this concern and give us the opportunity to explain our contribution better. While our method is intentionally simple, as the reviewer acknowledged, we believe it addresses a significant gap in the research community. Rather than treating model distillation as the straightforward transfer of knowledge from a single teacher to a student, our work reframes the problem within a more complex and heterogeneous landscape. Specifically, we focus on optimizing sampling strategies from an ensemble of teachers to target specific parts of the data distribution.
>
> Multilingual settings are particularly well-suited for this approach due to the distinct boundaries between languages compared to task-based distinctions. Through extensive experiments across 23 languages and 9 models, we demonstrated that arbitrage techniques enhance performance in scenarios where no single teacher model is state-of-the-art across all tasks. Additionally, we introduced and compared multiple routing strategies tailored to this setting, providing a novel perspective on multilingual synthetic instruction data generation.
>
> Our research takes an ecosystem-level view, emphasizing collaboration among multiple teacher models rather than relying on a single oracle. We hope the reviewer will recognize that this broader perspective, combined with our experimental results, highlights the significance of our contributions to advancing multilingual model training and synthetic data generation.

---

> ### Author Response · Authors · 2024-11-21
> **Reward-based and learned routing strategies**
>
> > **Also, the method, training a router to pick the best teacher for each sample, is quite straightforward, while the optimality of which cannot be guaranteed. Actually, the result that "reward-based routing" (using the score from an open reward model to choose the best from all the teachers' responses) outperforms "learned routing" (using a trained router) shows that the trained router doesn't always choose the teacher with the highest reward for that sample. Furthermore, the reward model score can also carry bias and error, bringing error accumulation in the process, but no experiment in this paper can tell whether the teacher with the highest reward is the ground-truth-best teacher for that sample (reviewed by human experts, etc.).**
>
> We would like to address the concerns raised regarding the reward-based routing and learned routing strategies.
>
> Reward models are designed to evaluate the quality of an LLM's outputs by assigning a scalar score (reward) to each response. This score reflects how well the response aligns with human preferences or predefined criteria. In our work, we use the reward model as an approximation of human preferences to select the best teacher for each sample. While the optimality of this approach cannot be guaranteed, our extensive experimental results demonstrate that reward-based routing significantly improves generative performance, textual quality, and deterministic task performance across 23 languages. These results demonstrate that the reward model is an effective tool for guiding the routing.
>
> Secondly, we would like to explain the learned router as discussed in Section 2.1 in the paper. We have observed that reward-based routing is a very efficient strategy to improve model performance, but it requires generating outputs from all teacher models for every sample, which can be computationally expensive. To mitigate this, we proposed training a smaller model (mT5-base with 580M parameters or Gemma2-2B) as a router. This learned router predicts the reward for a given prompt based on historical routing trends and scores produced by the reward model, allowing it to determine the most suitable teacher without requiring multiple generations. Since the learned router predicts the reward for each new prompt instead of directly assigning rewards to them, it is anticipated that its performance may not fully match reward-based routing. However, our experiments show that it serves as a good approximation and offers a practical trade-off between efficiency and performance. While it may not always select the teacher with the highest reward for each sample, it still achieves notable improvements and provides a scalable alternative in resource-constrained scenarios.

---

> ### Author Response · Authors · 2024-11-21
> **Multilingual arbitrage vs Random routing**
>
> > **The paper doesn't tell whether the "multilingual arbitrage" method is better than simply mixing the responses from multiple teacher models. Though there is a experiment comparing the reward-based routing method with the "single 'oracle' teacher" method (which is merely using the teacher model with the highest average performance), it is not enough**
>
> In our paper, we compare the multilingual arbitrage strategies not only with the single "oracle" teacher method but also with random routing, which effectively serves as a baseline for mixing responses from multiple teacher models. Random routing assigns each prompt to a teacher model randomly, simulating an approach where no specific routing strategy is applied. This comparison directly addresses the question of whether multilingual arbitrage is better than simply mixing responses from multiple teachers. Our results consistently show that multilingual arbitrage outperforms random routing across all metrics, demonstrating the effectiveness of our approach in leveraging teacher models more strategically rather than relying on random or naive mixing.

---

> ### Author Response · Authors · 2024-11-21
> **Clarification on experiment setting**
>
> > **Some of the experiment settings are questionable, for example:
> Aya-23-8B is used as both the student model and one of the teacher models
> Part of the Dolly-15k dataset is used in both the router training set and the system testing set**
>
> We selected the Aya 23 8B model as both a student and one of the teacher models due to its state-of-the-art multilingual capabilities relative to its size. While it is common practice to use a model’s own generations to enhance performance through self-distillation [2,3], it’s important to clarify that Aya 23 8B is not the sole teacher model used for synthetic data generation in our experiments.
>
> As shown in Figure 6, approximately 32% of prompts in the base setting are routed to Aya 23 8B, reflecting its role among a pool of teacher models. In larger settings, where we employ nine different teacher models, even fewer prompts are routed to Aya 23 8B, further diversifying the sources of knowledge.
>
> Additionally, we use a completely distinct dataset for distillation that has never been used for training or fine-tuning the student model. This ensures that the student model can still learn from its own generations, when prompts are routed to Aya 23 as the teacher, as well as from those of other teacher models, rather than simply reinforcing pre-existing knowledge.
>
> Regarding the evaluation, the subset of the Dolly-15k training dataset used for training the router did not overlap with the held-out testing data that is used to create Dolly200 eval set [3]. We appreciate the reviewer pointing out any potential ambiguity and we hope this clarification addresses your concerns.
>
>
> [1] Wang, Yizhong, et al. "Self-instruct: Aligning language models with self-generated instructions." arXiv preprint arXiv:2212.10560 (2022).
>
> [2] Yang, Zhaorui, et al. "Self-distillation bridges distribution gap in language model fine-tuning." arXiv preprint arXiv:2402.13669 (2024).
>
> [3] Üstün et al. "Aya model: An instruction finetuned open-access multilingual language model." ACL 2024.

---

> ### Author Response · Authors · 2024-11-21
> **Multilingual method and evaluation**
>
> > **The title and abstract of this paper put "multilingual" as a key point. However, throughout the methodology part, no specific design for the multilingual problem is proposed. Also, in the experment setting part, only 3 discriminative tasks are metioned to test the multilingual capacity of the trained student model, which is not satisfactory. In fact, the routing method can be applied to much more generation tasks, as long as there is need to distil open LLMs into a smaller model**
>
> While synthetic data has shown significant promise in addressing data scarcity and improving model performance, much of the focus has been on English. This gap highlights the importance of exploring multilingual synthetic data generation techniques to ensure more equitable advancements across diverse linguistic contexts. We completely agree that our routing method is highly flexible and applicable to a wide range of generation tasks, but our main objective in this work is to address this gap, so we proposed this strategic sampling for multilingual setting.
>
> For evaluation, we went beyond the three multilingual discriminative tasks (XNLI, XStoryCloze, XCOPA), which assess natural language inference, commonsense reasoning, and sentence completion. To further measure generative capabilities, we computed LLM-as-evaluator win rates on the Multilingual Dolly-200 Eval test set [3] that covers 23 languages. This evaluation set is specifically designed to assess multilingual generation capabilities, and we employed GPT-4 – one of the strongest multilingual models available as the judge. These results provide a comprehensive assessment of both discriminative and generative performance in multilingual contexts.
>
> We hope this clarification underscores how comprehensive our evaluation is, while also demonstrating the flexibility and effectiveness of our approach.
>
> [3] Üstün et al. "Aya model: An instruction finetuned open-access multilingual language model." ACL 2024.

---

> ### Author Response · Authors · 2024-11-21
> **KL loss for router training**
>
> >**Why is the router training loss the KL divergence instead of the normal cross-entropy? Since the goal of the router is to choose the single "best" teacher for a given sample, the cross-entropy loss against the highest-reward teacher of a given training sample seems more reasonable.**
>
> The goal of the router is to perform reward distillation, as described in lines 176 to 204 of the paper. We use the KL divergence loss because it is a standard choice in distillation tasks [4]. Unlike cross-entropy, which focuses solely on selecting the single "best" teacher, KL divergence allows the router to approximate the entire reward distribution over all teachers. This approach enables the router to capture richer information about the relative quality of different teachers for a given sample, rather than making a hard decision based only on the highest-reward teacher.
>
> [4] Lu et al., "Routing to the Expert: Efficient Reward-Guided Ensemble of Large Language Models." ACL, 2024.

---

> ### Author Response · Authors · 2024-11-24
> **Discussion period is coming to an end**
>
> We truly appreciate your thoughtful feedback and the time you’ve taken to evaluate our manuscript. As the discussion phase is nearing its conclusion, we kindly encourage you to review our responses to your comments. We hope our clarifications address your concerns, but please don’t hesitate to let us know if there are any remaining issues or additional details we can provide to assist in resolving them.

---

> ### Comment · Reviewer_su5q · 2024-11-25
> **Response to Authors**
>
> I appreciate the authors' detailed response. However, there are still some remaining questions and points I want to clarify:
>
> 1. For "Reward-based and learned routing strategies", my point is not only the optimality of the router cannot be guaranteed, but its routing performance is not even tested. Although the authors argue that there is significant increase in the final performance of the student model, it cannot justify this lack of evaluation of the router model.
>
> 2. For "Multilingual arbitrage vs Random routing", what I meant by "mixing the responses" is not "random routing", but using all the models' responses for each of the prompts. Let's say there are M models and N prompts, what I mean is to use the M*N responses to train the student model, and see whether the outcome is surpassed by the proposed methods, e.g. reward-based routing, given that their cost of calling teacher models are the same.
>
> 3. For "Clarification on experiment setting", thank you for clarifying the experiment settings. However, the self-distillation of Aya-23-8B is still questionable, because self-distillation is more than just using the model's own output as the SFT target, but requires special templates or procedures (as mentioned in [1-2] as you cited). Also, if you must use Aya-23-8B, there should be some evaluation results that demonstrate Aya-23-8B is more suitable than many other state-of-the-art open LLMs in 7-9B parameter sizes.
>
> 4. For "Multilingual method and evaluation", thank you for reminding me the results of Dolly-200. However, given there are only 100 samples, it also cannot represent the overall multilingual performance, especially given that there are plenty of novel multilingual benchmarks off the shelf.
>
> 5. For "KL loss for router training", thank you for the clarification of using KL loss instead of cross-entropy. However, the method in the cited paper [4] seems similar to the current paper, that is, training a reward-based router to assign prompts to the most suitable LLM. This again questions the novelty and contribution of this paper.

---

> ### Author Response · Authors · 2024-11-25
>
> Thank you for taking the time to review our responses and asking for more clarification. Below we will do our best to clarify these points further.
>
> 1. We compare the performance of all router models with the baselines—both random routing and single-model teachers -- for discriminative and generative tasks, as well as for textual quality evaluations throughout the paper, and we demonstrate consistent improvement. Reward models are evaluated separately on the RewardBench Leaderboard (https://huggingface.co/spaces/allenai/reward-bench), particularly in terms of how well they align with human preferences, and we selected one of the best-performing reward models in a multilingual context. Unfortunately, building a ground truth dataset annotated by human experts to identify the best completion is very costly, especially in large-scale multilingual settings where we need human experts native to 23 languages; thus, this was the most optimal option. We would be happy to consider any suggestions you may have for measuring reward-model success beyond relying on expert human annotator choices.
>
>
> 2. Sorry for the misunderstanding. We have this comparison as we already responded to Reviewer [**b94N**] and Reviewer [**Ee5t**]. This baseline is called “Full Budget” in the table below. In the submission, we have not added these results to the paper, since the data size in that variant is the number of teacher models times more than the other approaches. Here, we present the results, and we will also add them to the Appendix in the paper in revised version. Across all languages, reward-based routing (multilingual arbitrage) consistently achieved higher win rates.
>
> | Language | Reward Based Routing | Full Budget(All Completions) | Tie |
> |:-------------:|:-------------:|:-------------:|:-------------:|
> | English | 54.0 | 31.5 | 14.5 |
> | German | 47.5 | 33.5 | 19.0 |
> | French | 50.0 | 34.0 | 16.0 |
> | Arabic | 46.5 | 34.5 | 19.0 |
> | Chinese | 51.0 | 39.0 | 10.0 |
> | Turkish | 54.5 | 27.5 | 18.0 |
> | Ukrainian | 45.0 | 34.0 | 21.0 |
>
>
> 3. What specific special templates or procedures are you referring to? All models in the pool require special templates for data generation, and we adhered to these templates. We included Aya 23-8B as one of the teacher models because of its strong multilingual performance, particularly in non-mainstream languages such as Arabic, Turkish, and Ukrainian. As discussed in Section 3.1, even Llama 3 and Gemma 2 do not explicitly claim multilingual support (although they are often used by multilingual users more than models explicitly designed for multiple languages). Therefore, we needed a model that explicitly claims to perform well in such languages within that context.
>
>
> 4. Dolly-200 has 200 samples per language and in our larger scale setting the models are evaluated on 23 languages - that makes the generative evaluation on 4600 samples. With the suggestion of Reviewer [**87gS**], we ran additional experiments on Aya evaluation suite (https://huggingface.co/datasets/CohereForAI/aya_evaluation_suite),  where prompts were post-edited by human annotators after translation. This ensures that the dataset is free from any potential biases introduced by automated translation models. Below, we present results in the same setting as Figure 2 for Arabic, Chinese, Portuguese, and Turkish:
>
> | Language | Full Arbitrage | Gemma2-9B-IT | Tie |
> |:-------------:|:-------------:|:-------------:|:-------------:|
> | ar | 52.5 | 37.8 | 9.7 |
> | pt | 46.7 | 44.2 | 9.1 |
> | tr | 49.1 | 42.6 | 8.3 |
> | zh | 45.2 | 43.9 | 10.9 |
>
> These results demonstrate that our model performs effectively on human-edited datasets as well, confirming that our findings hold across both translated and human-edited evaluation sets. We will include the results on the whole evaluation set in the revised version of the paper.
>
> Additionally, discriminative task results on XNLI, XStoryCloze, and XCOPA are already presented. We welcome suggestions for further benchmarks to evaluate our models even more comprehensively.

---

> ### Author Response · Authors · 2024-11-25
> **Contribution of the paper**
>
> We apologize for any repetition, but we want to clarify that the main contribution of this paper is not the reward-based router. Instead, our primary contribution lies in rethinking model distillation as a complex and heterogeneous process, rather than a straightforward transfer of knowledge from a single teacher to a student. Specifically, we focus on optimizing sampling strategies from an ensemble of teachers to target specific parts of the data distribution in a multilingual setting, an area that has not been studied before. In this context, reward-based routing serves these objectives.
>
> Through extensive experiments across 23 languages and 9 models, we demonstrated that arbitrage techniques enhance performance in scenarios where no single teacher model is state-of-the-art across all tasks. Additionally, we introduced and compared multiple routing strategies tailored to this setting, providing a novel perspective on multilingual synthetic instruction data generation. Our research focuses on enhancing multilingual model performance through collaboration among multiple teacher models, rather than depending on a single oracle. Our experimental results highlight the importance of our contributions to advancing multilingual model training and synthetic data generation.
>
> We hope that our responses adequately address your concerns. Should you have any additional questions or require further clarification, we would be more than happy to discuss them and provide any necessary adjustments to improve the paper.

---

> ### Author Response · Authors · 2024-11-28
> **Paper Rating change**
>
> Dear Reviewer [**su5q**],
>
> We noticed that the rating for our paper was decreased from 5 to 3 without prior notice, despite our efforts to address all concerns and respond to the questions raised. We would greatly appreciate it if you could provide further insight into the reasons for this change. Understanding your perspective will be invaluable for us to improve our work.

---

> > ### Comment · Reviewer_su5q · 2024-12-01
> > **Response to Rating Change**
> >
> > I adjusted my score based on the authors' response during the rebuttal process, where my major concerns are raised but not properly addressed.

---

### Official Review · Reviewer_Ee5t · 2024-11-06

**Soundness:** 3
**Presentation:** 3
**Contribution:** 3
**Rating:** 8
**Confidence:** 3

**Summary:**

This paper looks at ways to leverage multiple teacher models for a student model to improve multilingual performance on a student model (Aya-23). Overall, this is an interesting idea and the experiments justify most of the claims. However, there are a few things that I would have liked to see in the paper.

One thing that wasn’t abundantly clear to me in the paper is that a prompt must only be routed to one expert? Why must this be the case? Is there a fixed budget? I would assume that using more than one model for each prompt would actually improve performance? Probably not all, but at least more than one. Assuming a fixed budget, I think the question then becomes, do we get more by using the same prompt and going through multiple models, or do we do better by getting more prompts and diversity in the data that way. I may be missing something here, but I think that is a better baseline.

However, overall, the idea of strategic sampling seems like a great idea.

Also, I don’t like the name arbitrage. This implies that the same thing is bought and sold (here I guess it means language data). Normally you can think of this as a cyclical graph where you are trying to find negative cycles (like Bellman-Ford algorithm). This is more of an efficiency routing algorithm, so the name doesn’t fit.


I’d also like to see results in an appendix broken down by language. Most of the results are averaged across all languages. What happens when you break it down? For instance, Table 2 is averaged across languages. It would be nice to see this broken down in the appendix. Table 8 A.7 is also averaged across 7 languages. This is already in appendix. It could be expanded.

**Strengths:**

A really interesting idea about strategic routing.

Multilingual tasks that are often overlooked.

**Weaknesses:**

Baseline in prompt.

Name "arbitrage".

**Questions:**

What are individual language scores?

What happens if the student model sees all models for a prompt? Or a subset as of all models?

---

> ### Author Response · Authors · 2024-11-20
> **Using more than one model for each prompt**
>
> We sincerely appreciate reviewer **Ee5t**'s acknowledgment of strategic sampling as an "interesting idea" and their recognition that our experiments "justify" most of our claims. We aim to address the reviewer’s concerns in detail below.
>
> >**One thing that wasn’t abundantly clear to me in the paper is that a prompt must only be routed to one expert? Why must this be the case? Is there a fixed budget? I would assume that using more than one model for each prompt would actually improve performance? Probably not all, but at least more than one. Assuming a fixed budget, I think the question then becomes, do we get more by using the same prompt and going through multiple models, or do we do better by getting more prompts and diversity in the data that way. I may be missing something here, but I think that is a better baseline.**
>
> Thank you for raising this point. While using multiple models per prompt could potentially improve performance, our experimental setup was specifically designed to address a different question: For a given set of prompts, is it better to rely on outputs from a single model or strategically sample outputs from a pool of models? To ensure fair comparisons, we kept the compute budget per prompt constant across all variants. Allowing multiple completions for a single prompt or chaining models would violate this constraint and make comparisons with the standard approach (sampling from a single model) unfair.
> That said, we have conducted this comparison. As we mentioned in our response to reviewer **b94N**, we did not include these results in the main paper because this variant requires significantly more data—proportional to the number of teacher models—than the other approaches. However, we will include these results in the Appendix of the revised paper. Notably, across all languages, reward-based routing (multilingual arbitrage) consistently achieved higher win rates.
>
> | Language | Reward Based Routing | Full Budget(All Completions) | Tie |
> |:-------------:|:-------------:|:-------------:|:-------------:|
> | English | 54.0 | 31.5 | 14.5 |
> | German | 47.5 | 33.5 | 19.0 |
> | French | 50.0 | 34.0 | 16.0 |
> | Arabic | 46.5 | 34.5 | 19.0 |
> | Chinese | 51.0 | 39.0 | 10.0 |
> | Turkish | 54.5 | 27.5 | 18.0 |
> | Ukrainian | 45.0 | 34.0 | 21.0 |
>
>
> > **What happens if the student model sees all models for a prompt? Or a subset as of all models?**
>
> We believe our response above, regarding the use of all teacher models per prompt addresses this question as well. As shown in the table above, reward-based routing consistently outperformed using generations from all teacher models (Full Budget) across all languages. This demonstrates that selectively routing prompts to specific models is more effective than utilizing all models for each prompt. As promised, we will include this result in the revised version of the paper.

---

> > ### Comment · Reviewer_Ee5t · 2024-12-03
> >
> > Thank you for clarifying "compute budget per prompt constant across all variants". I think this needs to be specifically addressed in the paper - and the results here will accomplish that.

---

> ### Author Response · Authors · 2024-11-20
> **Name: multilingual arbitrage**
>
> >**Also, I don’t like the name arbitrage. This implies that the same thing is bought and sold (here I guess it means language data).
> Normally you can think of this as a cyclical graph where you are trying to find negative cycles (like Bellman-Ford algorithm). This is more of an efficiency routing algorithm, so the name doesn’t fit.**
>
> We thank reviewer **Ee5t** for raising this concern and allowing us to explain our naming convention. We chose the term "arbitrage" because our routing method is designed to exploit performance variations across model pools during generation. In this context, the "asset" being optimized is the sampling compute, as we prioritize one teacher's generation over another to maximize performance.
>
> Additionally, the term aligns with our use of an "arbiter" -- a reward model or router trained to evaluate and select the optimal generation. The title "multilingual arbitrage" reflects this strategy in the multilingual domain, where natural differences in model strengths and performances are particularly pronounced.

---

> > ### Comment · Reviewer_Ee5t · 2024-12-03
> >
> > Thanks for clarifying your argument for the naming. However, I think "arbitrate" or "arbitration" are better derivatives of "arbiter" than "arbitrage" since it is an overloaded term in graph theory.

---

> > > ### Comment · Reviewer_Ee5t · 2024-12-03
> > > **Please change "arbitrage"**
> > >
> > > Honestly, this is the biggest concern that I have left with the paper. I really do not like the term "arbitrage" despite the derivate from "arbiter". Other definitions such as "arbitrate" or "arbitration" are much better. I considered not upping my score due to the name as I think it will confuse people in the community.

---

> ### Author Response · Authors · 2024-11-20
> **Individual language scores**
>
> > **What are individual language scores?**
>
> Thank you for the suggestion! We will include the individual language scores in the Appendix of the revised version. These scores are available for all cases discussed in the paper. Here, we are providing the individual language scores for the results shown in Figure 2, which compares reward-based routing-trained students with the best-performing state-of-the-art model in our experiments, Gemma2-9B-IT.
>
> | Language | Full Arbitrage | Gemma2-9B-IT | Tie |
> |:-------------:|:-------------:|:-------------:|:-------------:|
> | ar | 57.5 | 36.0 | 6.5 |
> | cs | 50.5 | 42.5 | 7.0 |
> | de | 50.0 | 46.0 | 4.0 |
> | el | 57.0 | 37.5 | 5.5 |
> | en | 37.0 | 57.0 | 6.0 |
> | es | 41.0 | 52.5 | 6.5 |
> | fa | 57.0 | 35.5 | 7.5 |
> | fr | 35.0 | 55.5 | 9.5 |
> | he | 65.0 | 29.0 | 6.0 |
> | hi | 50.0 | 44.0 | 6.0 |
> | id | 53.0 | 42.0 | 5.0 |
> | it | 49.5 | 42.0 | 8.5 |
> | ja | 44.0 | 45.5 | 10.5 |
> | ko | 53.0 | 34.5 | 12.5 |
> | nl | 49.0 | 46.0 | 5.0 |
> | pl | 49.0 | 45.0 | 6.0 |
> | pt | 46.5 | 45.5 | 8.0 |
> | ro | 58.0 | 37.5 | 4.5 |
> | ru | 50.5 | 44.5 | 5.0 |
> | tr | 47.5 | 45.0 | 7.5 |
> | uk | 48.5 | 44.5 | 7.0 |
> | vi | 53.5 | 42.5 | 4.0 |
> | zh | 50.5 | 44.5 | 5.0 |

---

> > ### Comment · Reviewer_Ee5t · 2024-12-03
> >
> > Thank you for adding this. It will be useful to the paper.

---

> ### Author Response · Authors · 2024-11-24
> **Discussion period is coming to an end**
>
> We sincerely thank you for your constructive comments and evaluation of our manuscript. As the discussion phase will be closed soon, we kindly ask you to take a look at our responses and reevaluate our work based on our clarifications. Please let us know whether our response addresses your concerns or whether there is any further detail we can provide to help address these concerns.
>
> Thank you again for dedicating your time to reviewing our paper.

---

> ### Author Response · Authors · 2024-11-27
>
> Dear Reviewer Ee5t,
>
> As the discussion phase will be closed soon, we kindly ask you to review our responses and reevaluate our work based on our clarifications. Please let us know whether our response addresses your concerns or whether there is any further detail we can provide to help address these concerns. We would be happy to discuss more if you have any questions remained.
>
> Thank you!

---

### Official Review · Reviewer_b94N · 2024-11-09

**Soundness:** 3
**Presentation:** 3
**Contribution:** 2
**Rating:** 8
**Confidence:** 3

**Summary:**

This work introduces the concept of multilingual arbitrage to generate better synthetic data by using multiple teachers, each of which is specialized in their corresponding language. This reduces the bias propagation and model collapse from using a single teacher model, which is especially prevalent in multilingual contexts. The authors introduce different kind of routing strategies, namely

1)Fixed-routing with predefined set of expert teachers

2)Reward-based routing in which all the teacher completions are considered and only the top ranked teacher completion is used to train the student model

3)Learned routing to address the disadvantage of the reward based routing where all teacher completions have to be generated. In this strategy, the model learns to choose the best teacher based on the given prompt and uses the completion of that teacher to train the student model

The above strategies have been tested across 15 languages and 9 SOTA multilingual models. The experiments show that reward-based routing technique achieved significant improvement over the best single-teacher model. They also performed ablation against random routing and showed that all three strategies significantly improve the performance against random routing. Scaling up arbitrage setup also improved the win-rate significantly against SOTA multilingual models like Gemma, Llama 3.1, and Mistral v0.3. The authors also assess the generated text's verbosity, readability, and diversity of the student model, finding that multilingual arbitrage improves all these characteristics.

**Strengths:**

1) The authors introduces a unique "multilingual arbitrage" approach that routes samples to the most suitable model for each language, thereby creating better student models than from a single best teacher

2) The authors have done a comprehensive evaluation of the different types of routing strategies, including testing for textual quality metrics like readability, verbosity, and lexical diversity, contributing insights into how multilingual arbitrage affects text quality

3) Win-rate comparisons across different resourced languages show that medium resourced languages like Turkish and Ukrainian experience a higher gain than high resource languages like English, which helps in addressing a critical gap of improving performance for languages other than high resourced ones

4) The results in Figure 7 indicate that translation is the least effective method for synthetic data generation, as even random routing performs better. This result helps in generating synthetic data which are suited to many contexts like tasks requiring cultural cues or other tasks like XNLI which are translated from English

**Weaknesses:**

1)The authors mention that the multilingual arbitrage method is created to not rely on a single model which creates bias and model collapse but do not show any experiments on how this is avoided

2)The language families considered is limited, with significant gaps, particularly in Indian and other South Asian languages, which limits the generalizability of the multilingual arbitrage method

3)The authors mainly rely on quantitative metrics with limited qualitative analysis of generated text in terms of cultural accuracy, idiomatic expressions, or semantic consistency, which is crucial in multilingual contexts

**Questions:**

1)How are the reward models used for the "LLM-as-a-judge" and the reward-based routing strategy different, and what criteria are used to evaluate their effectiveness?

2)Have the authors tested using all the completions  to finetune the student model to verify whether the reward-based routing is truly competitive?

3)Why is only a single English model used for English-only prompts in the translation setting? Would a comparison with multiple English models provide more robust insights?

4)Why is Turkish categorized within the East-Asian language cluster?

5)In the basic set, the teacher and student models are essentially the same size. What is the rationale for performing distillation in this case?

6)The improvement achieved by learned routing is less than that of fixed routing. Are there any insights on why learned routing is less performant?

7)The figures and the corresponding discussion in Figure 5 seem inconsistent. Could the authors clarify these discrepancies?

---

> ### Author Response · Authors · 2024-11-20
> **Bias evaluation**
>
> We appreciate reviewer **b94N** for their feedback and acknowledging the effort we have put into creating a ``comprehensive'' evaluation of the different routing strategies, and how our method addresses the “critical gap of improving performance for languages other than high resourced ones”.
>
> > **The authors mention that the multilingual arbitrage method is created to not rely on a single model which creates bias and model collapse but do not show any experiments on how this is avoided**
>
> We do not explicitly measure the bias in our evaluations, but the winrate results where we compare arbitrage approaches to single teacher models, presented in Figure 4 and Figure 5, and the detailed textual characteristics analysis presented in Appendix A.6 shows that multilingual arbitrage methods avoid model collapse. We see substantial improvements in the readability of the generations (measured by Gunning Fog index and Rix) and textual lexical diversity as shown in Table 9. The number of tokens per generation also increases, along with readability and lexical diversity.

---

> ### Author Response · Authors · 2024-11-20
> **Generalizability of the multilingual arbitrage method**
>
> >**The language families considered is limited, with significant gaps, particularly in Indian and other South Asian languages, which limits the generalizability of the multilingual arbitrage method**
>
> As detailed in Section 3.3, we conducted two sets of experiments:
> 1) **Basic Set**, where synthetic data is generated in seven languages: Arabic, Chinese, English, French, German, Turkish, and Ukrainian.
> 2) **Larger Scale**, where synthetic data is generated in 23 languages. This includes all the languages from the Basic Set, plus Dutch, Czech, Greek, Spanish, Persian, Hebrew, Hindi, Indonesian, Italian, Japanese, Korean, Polish, Portuguese, Russian, and Vietnamese.
>
> These languages span seven language families and nine scripts. Detailed lineage information for these languages can be found in Table 6 in Appendix A.2. The results of the larger scale experiments are presented in Figure 2 and discussed thoroughly in Section 4.1.
> Regarding the Asian languages included in these experiments, we generated data for **Hindi, Korean, Vietnamese, and Japanese**. We believe that this broad language coverage and the improvements observed across these languages demonstrate the generalizability of the multilingual arbitrage method.

---

> ### Author Response · Authors · 2024-11-20
> **Multilingual Evaluations**
>
> > **The authors mainly rely on quantitative metrics with limited qualitative analysis of generated text in terms of cultural accuracy, idiomatic expressions, or semantic consistency, which is crucial in multilingual contexts**
>
> Thank you for your suggestion regarding cultural accuracy and idiomatic expressions. While we recognize the importance of these elements in multilingual contexts, our primary focus in this work is to evaluate the general performance of the model using a combination of quantitative, generative, and qualitative metrics. To assess text quality, we employed readability measures such as Gunning Fog, MLTD, and Rix, which offer a quantitative evaluation of linguistic complexity and fluency. Additionally, we evaluated the model's performance on deterministic benchmarks like XCOPA, XNLI, and XStoryCloze – widely regarded for testing reasoning, inference, and story comprehension across multiple languages. Furthermore, we used LLM-as-an-evaluator win rates with GPT-4 (one of the strongest models) to measure our generative performance on the Multilingual Dolly-200 Eval test [1] , which is currently one of the most widely adopted approaches for evaluating generative performance.
> Expanding our evaluation to include more detailed assessments of cultural nuances and idiomatic usage would certainly enrich our understanding of the model’s capabilities in diverse linguistic environments, but this is very hard to achieve without native, professional human annotators. But, we would be happy to explore these evaluations in future work.
>
> [1] Üstün et al. "Aya model: An instruction finetuned open-access multilingual language model." ACL 2024.

---

> ### Author Response · Authors · 2024-11-20
> **Reward model and LLM-as-an-evaluator models**
>
> >**How are the reward models used for the “LLM-as-a-judge” and the reward-based routing strategy different, and what criteria are used to evaluate their effectiveness?**
>
> In section 2.1, we mention that the rewards model used for the reward-based routing strategy (Cohere May 2024) and the LLM-as-a-Judge (GPT-4-Turbo) are different. We ensure this due to known biases that occur when using the same models for both. We used (Cohere May 2024) because it was one of the top models in RewardBench Leaderboard (https://huggingface.co/spaces/allenai/reward-bench) and multilingual at the same time and GPT-4-Turbo was one the top model in most of the benchmarks.

---

> ### Author Response · Authors · 2024-11-20
> **All completions vs Multilingual Arbitrage**
>
> > **Have the authors tested using all the completions to finetune the student model to verify whether the reward-based routing is truly competitive?**
>
> Yes, we have tested that, but we have not added these results to the paper, since the data size in that variant is the number of teacher models times more than the other approaches. Here, we present the results, and we will also add them to the Appendix in the paper in revised version. Across all languages, reward-based routing (multilingual arbitrage) consistently achieved higher win rates.
>
> | Language | Reward Based Routing | Full Budget(All Completions) | Tie |
> |:-------------:|:-------------:|:-------------:|:-------------:|
> | English | 54.0 | 31.5 | 14.5 |
> | German | 47.5 | 33.5 | 19.0 |
> | French | 50.0 | 34.0 | 16.0 |
> | Arabic | 46.5 | 34.5 | 19.0 |
> | Chinese | 51.0 | 39.0 | 10.0 |
> | Turkish | 54.5 | 27.5 | 18.0 |
> | Ukrainian | 45.0 | 34.0 | 21.0 |

---

> ### Author Response · Authors · 2024-11-20
> **Why is only a single English model used?**
>
> >**Why is only a single English model used for English-only prompts in the translation setting? Would a comparison with multiple English models provide more robust insights?**
>
> In the translation setting, we aim to answer the following – “does generating synthetic data in the target language outperform translating the best English completions to the target language?” In the typical scenarios, the synthetic data is generated from a single model and then translated into different languages. The upper bound for that is to use the strongest model in English and translate the data from that model’s generations. Our experiments show that Llama3 is the strongest model in English (Figure 6 shows that 60% of the English prompts are routed to Llama3).  Given this evidence, we opted to use Llama3's completions for translation to maximize performance.

---

> ### Author Response · Authors · 2024-11-20
> **Turkish is categorized within the East-Asian language cluster**
>
> > **Why is Turkish categorized within the East-Asian language cluster?**
>
> While we recognize that Turkish would ideally be in a Turkic language family cluster, since we don’t have other languages to create that cluster, we chose to include it in the East-Asian geo-cluster for our experiments. The historical origins of the Turkic languages, which are believed to have originated in a region between Mongolia and Northwest China (East Asia) (https://en.wikipedia.org/wiki/Turkic_languages). Given that our geo-cluster experiment was based on broad geographical origins rather than strict linguistic classification, Turkish's East Asian roots made it a reasonable fit for this cluster.

---

> ### Author Response · Authors · 2024-11-20
> **Size of student and teacher models**
>
> > **In the basic set, the teacher and student models are essentially the same size. What is the rationale for performing distillation in this case?**
>
> Thank you for highlighting this point. We selected the Aya 23 8B model as both a student and one of the teacher models due to its state-of-the-art multilingual capabilities relative to its size. While it is common practice to use a model’s own generations to enhance performance through self-distillation [2,3], it’s important to clarify that Aya 23 8B is not the sole teacher model used for synthetic data generation in our experiments.
>
> As shown in Figure 6, approximately 32% of prompts in the base setting are routed to Aya 23 8B, reflecting its role among a pool of teacher models. In larger settings, where we employ nine different teacher models, even fewer prompts are routed to Aya 23 8B, further diversifying the sources of knowledge.
>
> Additionally, we use a completely distinct dataset for distillation that has never been used for training or fine-tuning the student model. This ensures that the student model can still learn from its own generations, when prompts are routed to Aya 23 as the teacher, as well as from those of other teacher models, rather than simply reinforcing pre-existing knowledge.
>
> [2] Wang, Yizhong, et al. "Self-instruct: Aligning language models with self-generated instructions." arXiv preprint arXiv:2212.10560 (2022).
>
> [3] Yang, Zhaorui, et al. "Self-distillation bridges distribution gap in language model fine-tuning." arXiv preprint arXiv:2402.13669 (2024).

---

> ### Author Response · Authors · 2024-11-20
> **Fixed Routing vs Learned Routing**
>
> >**The improvement achieved by learned routing is less than that of fixed routing. Are there any insights on why learned routing is less performant?**
>
> Thank you for giving us a chance to clarify this point. Fixed routing is the setting where we have prior information about which model works best for a particular language because we fine-tuned the geo-cluster models only for certain family of languages  to exploit geographic and linguistic similarities within a language cluster. For example Germanic geo-cluster is specialized for German and Dutch, Slavic geo-cluster is specialized for Czech, Russian, Ukrainian, Polish etc. and we route the German prompts to Germanic geo-cluster and Ukrainian prompts to Slavic cluster for data generation. It is expected such routing with strong prior information to work better than a reward-based learned routing. But it is not realistic to have this strong prior information in all cases, so learned routing provides a more practical solution.

---

> ### Author Response · Authors · 2024-11-20
> **Clarification for Figure 5**
>
> >**The figures and the corresponding discussion in Figure 5 seem inconsistent. Could the authors clarify these discrepancies?**
>
> Of course we are happy to clarify this point. While Figure 5 shows the absolute win-rate scores for different resource(medium/high) languages when using different routing strategies (reward-based and learned) compared to single teacher models, we further discuss the relative performance gains in the discussion for an added nuance of improvement. The gains reported used the following formula: **(% winrate using routing - % winrate using single teacher) /  % winrate using single teacher**. We will clarify this in the text as well.

---

> ### Author Response · Authors · 2024-11-24
> **Discussion period is coming to an end**
>
> We sincerely thank you for your constructive comments and evaluation of our manuscript. As the discussion phase will be closed soon, we kindly ask you to take a look at our responses and reevaluate our work based on our clarifications. Please let us know whether our response addresses your concerns or whether there is any further detail we can provide to help address these concerns.
>
> Thank you again for dedicating your time to reviewing our paper.

---

> > ### Comment · Reviewer_b94N · 2024-11-25
> >
> > The authors have addressed my concerns, and I have updated my score accordingly

---

> ### Author Response · Authors · 2024-11-25
> **Thank you**
>
> Thank you for taking the time to review our responses and revisions, and for reconsidering your evaluation of our paper. We greatly appreciate your thoughtful and detailed feedback.

---

### Author Response · Authors · 2024-11-28
**Revised submission**

We sincerely thank all reviewers for their invaluable feedback. We have revised the paper and submitted the updated version. Based on the suggestions from Reviewers [**b94N**], [**su5q**] and [**Ee5t**], we included results for the variant using all completions to fine-tune the student model. These results are **presented in Appendix A.8: FULL BUDGET COMPARISON**. Following Reviewer [**Ee5t**]'s suggestion, we added language-specific win rates, which are **presented in Appendix A.9: LANGUAGE-SPECIFIC WIN RATES**. As pointed out by Reviewer [**87gS**], we corrected the citation, and **changed the main score reporting to absolute differences**, as suggested. These changes are highlighted in red in the revised submission.

To enhance the coverage of model evaluations, we have begun evaluating our models on the Aya evaluation suite (https://huggingface.co/datasets/CohereForAI/aya_evaluation_suite), where prompts are post-edited by human annotators after translation. This ensures that the dataset is free from potential biases introduced by automated translation models.
In this rebuttal, we present results in the same setting as Figure 2 for Arabic, Chinese, Portuguese, and Turkish:

| Language | Full Arbitrage | Gemma2-9B-IT | Tie |
|:-------------:|:-------------:|:-------------:|:-------------:|
| ar | 52.5 | 37.8 | 9.7 |
| pt | 46.7 | 44.2 | 9.1 |
| tr | 49.1 | 42.6 | 8.3 |
| zh | 45.2 | 43.9 | 10.9 |

These results demonstrate that our model performs effectively on human-edited datasets, confirming that our findings hold across both translated and human-edited evaluation sets. Once this evaluation is completed on the entire evaluation set, we will include them in the next version of the paper.

---

### Meta-Review · Area_Chair_X27n · 2024-12-26

**Metareview:**

This paper introduces the idea of “multilingual arbitrage” for synthetic data generation, leveraging multiple teacher LLMs instead of a single, monolithic teacher to train a smaller student model. The authors compared three routing strategies for selecting the teachers' output, showing that reward-based routing achieves the highest overall performance.

**Strengths** (1) The paper explored an interesting problem of using multiple teacher models to mitigate bias propagation in certain languages. (2) The evaluation is comprehensive. The experiments cover 15 languages and compared with various routing strategies.

**Weaknesses** (1) Some reviewers contend the core idea of using multiple teacher LLMs is straightforward, which questioned the novelty of the research contribution; (2) Reviewer mentioned while the authors claim that “multilingual arbitrage” alleviates bias or collapse, there is no direct experiment showing how single-teacher biases are reduced. (3) Some reviewers questioned about the overlapping issue of teacher/student and data leakage. (4) The confusion of the title using "arbitrage".

**Decision** This paper received mixed reviews, making it in a borderline region.

On one hand, the paper received two positive reviews with scores of 8, indicating the solid contributions from the reviewers.
However, the other two reviewers keep the rating negative. In both cases, there are concerns related to the method/experiment settings of reward-based routing, and the discussion does not conclude the rebuttal phase. Importantly, Reviewer su5q explicitly mentioned their question was not addressed and decreased the score, which led to concerns whether the paper was ready to be published at ICLR.

Therefore, my judgment leans toward rejection but I will leave SAC to make the final decision.

**Additional Comments On Reviewer Discussion:**

b94N questions are mainly in the experiment details, and the authors answered them properly, which helped address the original concerns and the score was raised accordingly.

The major discussion between Ee5t surrounds the term "arbitrage" which potentially causes additional confusion. The authors understand the concern and promise the change of the term.

su5q questioned about the experimental details with additional clarification questions. The author responded with additional results for part of the questions, but su5q was not satisfied with the response. Although su5q did not specifically mentioned the question, it may be due to the lack of evaluation of the reward models.

87gS summarized the discussion with the authors and noting the concerns on "RM and LLM-as-a-judge correlation", which 87gS thinks such exploitation is vulnerable to over-optimization to the metric.

---

### Decision · Program_Chairs · 2025-01-22

Reject